# Beta-cell specific *Insr* deletion promotes insulin hypersecretion and improves glucose tolerance prior to global insulin resistance

Insulin receptor (Insr) protein is present at higher levels in pancreatic β-cells than in most other tissues, but the consequences of β-cell insulin resistance remain enigmatic. Here, we use an *Ins1*^cre knock-in allele to delete *Insr* specifically in β-cells of both female and male mice. We compare experimental mice to *Ins1*^cre-containing littermate controls at multiple ages and on multiple diets. RNA-seq of purified recombined β-cells reveals transcriptomic consequences of *Insr* loss, which differ between female and male mice. Action potential and calcium oscillation frequencies are increased in *Insr* knockout β-cells from female, but not male mice, whereas only male βInsr^KO islets have reduced ATP-coupled oxygen consumption rate and reduced expression of genes involved in ATP synthesis. Female βInsr^KO and βInsr^HET mice exhibit elevated insulin release in ex vivo perifusion experiments, during hyperglycemic clamps, and following *i.p.* glucose challenge. Deletion of *Insr* does not alter β-cell area up to 9 months of age, nor does it impair hyperglycemia-induced proliferation. Based on our data, we adapt a mathematical model to include β-cell insulin resistance, which predicts that β-cell *Insr* knockout improves glucose tolerance depending on the degree of whole-body insulin resistance. Indeed, glucose tolerance is significantly improved in female βInsr^KO and βInsr^HET mice compared to controls at 9, 21 and 39 weeks, and also in insulin-sensitive 4-week old males. We observe no improved glucose tolerance in older male mice or in high fat diet-fed mice, corroborating the prediction that global insulin resistance obscures the effects of β-cell specific insulin resistance. The propensity for hyperinsulinemia is associated with mildly reduced fasting glucose and increased body weight. We further validate our main in vivo findings using an *Ins1*-CreERT transgenic line and find that male mice have improved glucose tolerance 4 weeks after tamoxifen-mediated *Insr* deletion. Collectively, our data show that β-cell insulin resistance in the form of reduced β-cell *Insr* contributes to hyperinsulinemia in the context of glucose stimulation, thereby improving glucose homeostasis in otherwise insulin sensitive sex, dietary and age contexts.

Type 2 diabetes is a multifactorial disease. Several cell types, most prominently pancreatic β-cells, are dysfunctional prior to and after diagnosis[1]. Hyperinsulinemia, insulin resistance, impaired fasting glucose, and impaired glucose tolerance can all be observed prior to the onset of frank diabetes[2], but the causal relationships between these factors remain incompletely understood[3]. Impaired insulin receptor (Insr) signaling is associated with obesity and often precedes the onset of overt type 2 diabetes, but it has been studied primarily in skeletal muscle, fat, and liver where it manifests differently[4]. Recent work in mice has established that β-cell specific insulin resistance can be observed early in the progression towards type 2 diabetes, when hyperinsulinemia is prominent, and independently of insulin resistance in other tissues[5]. The physiological consequences of reduced Insr in β-cells remain controversial.

It remains unresolved whether physiological insulin action on β-cells manifests as positive feedback to stimulate further insulin secretion, or negative feedback to inhibit its own release[6]. Human studies provide evidence for both possibilities. In vivo hyperinsulinemic-euglycemic clamps can reduce circulating C-peptide, a marker of endogenous insulin secretion[7,8]. In some studies, this inhibition was impaired in the obese state suggesting that systemic insulin resistance also extends to β-cells[8]. Bouche and colleagues replicated this result at basal glucose, but also found evidence that insulin can potentiate glucose-stimulated insulin secretion under specific conditions[9]. Others have shown that insulin can either stimulate or inhibit its own secretion depending on the metabolic context[10]. Administration of insulin to single β-cells in vitro increases intracellular calcium ($Ca^{2+}$)[11] and, in some studies, stimulates exocytosis[12]. However, $Ca^{2+}$ release from intracellular stores is not always sufficient to evoke insulin exocytosis. Studies in human β-cells did not detect robust exocytosis or C-peptide release in response to exogenous insulin despite observed intracellular $Ca^{2+}$ release[13].

Whether chronic deviations in autocrine insulin signaling affect β-cell development, survival, and adaptation is also controversial. Mice with chronically reduced insulin production have impaired β-cell expansion in the context of a high fat diet[14]. In vitro, physiologically relevant concentrations of insulin support the survival of both human and mouse β-cells[15]. We also reported that insulin is sufficient to increase proliferation of cultured primary mouse β-cells and that blocking insulin secretion with somatostatin blunts proliferation induced by hyperglycemia[16], and that the majority of glucose-dependent changes in gene expression in MIN6 cells are Insr-dependent[17]. However, hyperglycemia-induced β-cell proliferation has also been proposed to bypass Insr[18,19]. Thus, whether trophic signals from insulin and/or glucose are transmitted through Insr, Igf1r, or both receptors, remains unresolved.

To address the short-term and long-term consequences of eliminating Insr signaling in vivo, Kulkarni and colleagues crossed mice with floxed Insr alleles with mice expressing an Ins2 promoter driven Cre transgene[20]. Using this and related models, they reported that mice lacking β-cell Insr had profound glucose intolerance and frank diabetes in some cases, due to impaired glucose-stimulated insulin secretion, Glut2 loss, and insufficient β-cell mass[20–22]. Their Insr deficient mice failed to exhibit the compensatory increase in β-cell mass that accompanies a high fat diet[21]. Doubt was cast on these results when these Cre lines were subsequently shown to have off-target tissue effects owing to endogenous Ins2 expression in the brain and thymus[14,23–26]. More recently, Wang and colleagues studied the roles of β-cell Insr in utero and in adult mice using an inducible Ins1-CreER transgenic mouse model[27,28], but these studies are confounded by the presence of the human growth hormone (hGH) minigene[29], which necessitates the use of Cre-containing controls exclusively.

In the present study, we primarily used the constitutive Ins1[Cre] knock-in strain with robust and specific recombination in β-cells[30] to precisely reduce Insr signaling and define its consequences on glucose homeostasis. We validated our findings on glucose homeostasis using an additional β-cell specific model. Using this multi-factorial approach, we find clear evidence that Insr signaling plays a suppressive role on insulin secretion by modulating β-cell electrical excitability and that this effect is absent in conditions of global insulin resistance.

## Results

**Insr abundance in human islets and β-cells.** We initiated our studies by conducting an unbiased analysis of INSR abundance across tissues using publicly accessible data. Pancreatic islets had the second highest protein abundance of both isoforms of the INSR across a panel of 24 human tissues, as quantified by mass-spectrometry (Fig. 1a). These results, which are not complicated by the limitations associated with anti-INSR antibodies, suggest that human islets can have more INSR protein than "classical" insulin target tissues, including the liver and adipose. This also supports our previous observations illustrating that INSR are more abundant in β-cells relative to neighboring cells in the mouse pancreas[31]. Compilation of open source public single-cell RNA sequencing data from human islets demonstrated INSR mRNA in 62.4% β-cells, alongside other islet cell types (Fig. 1a, b). Clearly, β-cells have evolved for Insr-mediated autocrine signaling.

**β-cell specific Insr deletion with Ins1[Cre] mice.** We next sought to examine the function of the Insr, and the consequences of β-cell-specific insulin resistance/sensitivity, using an in vivo β-cell specific knockout mouse model. To limit recombination of the floxed Insr allele to pancreatic β-cells, we used a Cre allele knocked into the endogenous Ins1 locus which, unlike Ins2 promoters, drives specific expression in β-cells[30]. Experimental Insr[f/f];Ins1[cre/wt];nTnG (βInsr[KO]) and Insr[f/wt];Ins1[cre/wt];nTnG (βInsr[HET]) mice and littermate control Insr[wt/wt];Ins1[cre/wt];nTnG mice were generated using a breeding scheme that ensured consistency of the Cre donor parent (Fig. 1c). Cre-recombinase efficiency was assessed using the nuclear TdTomato-to-nuclear EGFP (nTnG) lineage trace reporter[32] and found to be robust on the Insr[f/f];Ins1[cre/wt];nTnG genetic background (Fig. 1d). We confirmed by Western blotting that Insr protein was almost completely absent from βInsr[KO] islets and partially reduced from βInsr[HET] islets (Fig. 1e). qPCR showed that Insr mRNA was not decreased in any of the 18 tissues examined, including the whole brain or hypothalamus; hence, we did not perform Western blots for other tissues (Fig. 1f). Together with other published data on Ins1[cre] mice, these findings strongly suggest that Insr deletion with Ins1[cre] is robust and sufficiently specific to pancreatic β-cells.

**Loss of β-cell Insr alters gene expression in purified β-cells.** To establish the baseline gene expression profile of our β-cell specific Insr knockout model, we performed an unbiased analysis of gene expression in ~100 FACS purified GFP-positive β-cells labeled with the nTnG reporter isolated from both female and male mice (Fig. 2a). We found that Insr mRNA expression was similar in female and male β-cells (Fig. 2b). We confirmed a similar reduction in Insr in female and male βInsr[KO] β-cells, with an intermediate phenotype found in the βInsr[HET] β-cells (Fig. 2b). We did not observe a compensatory change in Igf1r mRNA (Fig. 2b). After excluding samples with insufficient β-cell purity and analyzing across both sexes, RNA sequencing revealed significant differences in the expression of 12 genes between βInsr[KO] β-cells and wildtype β-cells (Fig. 2c, d and Fig. S1). However, when we analyzed female and male βInsr[KO] β-cells separately, we identified sex-specific gene

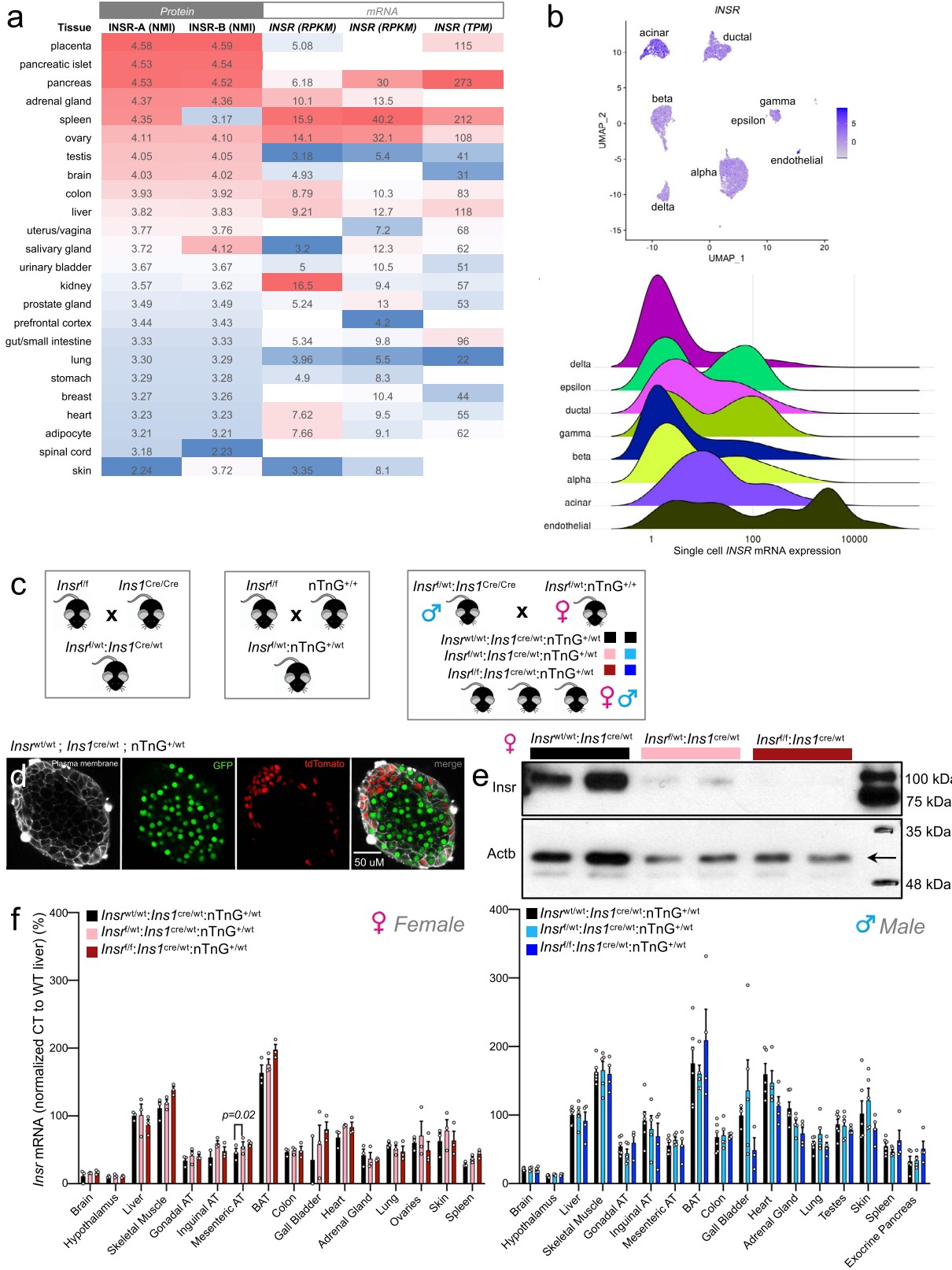

expression changes (Fig. 2d–f). In female-only analysis, five genes (including two pseudo-genes) were differentially expressed, while in male-only analysis, 64 genes were differentially expressed with an adjusted $p$ value of <0.05. At this cut-off, there was no overlap between the sex-specific gene expression patterns. Although our study was not designed or powered for direct comparisons between sexes, these data highlight the importance of considering both sexes separately.

**Loss of β-cell Insr increases β-cell excitability**. Insulin administration has been reported to open β-cell $K_{ATP}$ channels, mediating negative feedback on insulin secretion[33]. Lack of this

**Fig. 1 An animal model to examine the role of abundant Insr in β-cells. a** INSR protein (isoform A and B) and *INSR* mRNA expression across human tissues. *Left 3 columns* show INSR isoform A and B protein abundance expressed as normalized median intensity (NMI; www.proteomicsdb.org). *Right 3 columns* show total *INSR* mRNA expression in the same tissues (where available) extracted from three databases. *Left* to *right*, mRNA data from human proteome atlas (HPA) and genotype tissue expression project (GTEX) are shown in reads per kilobase million (RPKM). Data from the FAMTOMS database is shown as transcripts per kilobase million (TMP). High and low expression levels within a dataset are indicated by red and blue cells, respectively. **b** Human *INSR* expression in islet cell subtypes extracted from an integrated dataset of human single cell RNA seq data (see "Methods" and "Data availability" sections). Normalized expression levels are shown in UMAP space (*top*) and as a ridge plot on a log scale (*bottom*). The height of the ridge indicates the frequency of cells at a given expression level. **c** Breeding strategy for generating β-cell specific βInsr^KO, βInsr^HET, and littermate control Cre-only mice. **d** Robust Cre recombination verified by live cell imaging of the nTnG reporter allele (GFP = green; tdTomato = red) in an isolated islet from an *Ins1*^Cre/wt;nTnG mouse incubated with with CellMask™ Deep Red Plasma membrane stain (white). Similar recombination was observed across multiple independent experiments (*n* > 10). **e** Western blot of Insr protein in islets isolated from individual βInsr^KO (dark red), βInsr^HET (light pink), and littermate control mice (black). Similar results were observed in islets from individual mice (*n* = 5). **f** *Insr* mRNA expression across tissues in 16 week-old LFD-fed control (black bars), βInsr^HET (female = light pink bars, male = light blue bars) and βInsr^KO mice (female = dark red bars, male = dark blue bars) assessed by qPCR (*n* = tissues from 3 females, *n* = tissues 5–6 in males). Data were analysed by a mixed-effects model with correction for multiple comparisons using Dunnett's method. All data are presented as mean values +/− SEM. All animals/samples were handled in a strictly blinded manner. Source data are provided as Source Data file.

endogenous insulin action through Insr would therefore be expected to lead to β-cell hyper-excitability in our mouse model in the context of high glucose. Indeed, electrophysiological analysis of single β-cells from female βInsr^KO mice confirmed a significant increase in action potential firing frequency during glucose stimulation, when compared to control β-cells, with no differences in resting potential, firing threshold, or action potential height (Fig. 3a, b). The reversal potential was right-shifted in βInsr^KO β-cells, further suggesting reduced K^+ conductance (Fig. 3c). Hyper-excitability was not observed in *Insr* knockout β-cells from male mice, at the age we studied (Fig. 3d). We did not observe a statistically significant difference in depolarization induced exocytosis in single cells from either sex (Fig. 3e, f), suggesting that the late stages of insulin granule exocytosis are not altered under these conditions.

Next, we analyzed Ca^2+ responses to 15 mM glucose in thousands of Fura-2-loaded dispersed islet cells using an adaptation of our TraceCluster algorithm[34]. In agreement with the electrophysiological data, *Insr* knockout β-cells from female mice exhibited a significantly greater number of oscillation peaks within the glucose stimulation period compared to control cells (Fig. 3g, h and Fig. S2a). A similar increase in excitability was observed in βInsr^HET β-cells. This was not associated with significant differences in the intensity or gross localization of Glut2 protein using immunofluorescence imaging (Fig. S2b).

We examined mitochondrial function using the Seahorse bioanalyzer in the context of 10 mM glucose. In dispersed islet cells isolated from female mice, there were no significant differences between genotypes in oxygen consumption rate, with the exception of a reduced proton leak (Fig. 3i). In contrast, dispersed islet cells from both βInsr^KO and βInsr^HET males had a significant reduction in glucose-stimulated oxygen consumption rate compared to controls. Oligomycin injection revealed that this included a decrease in ATP-linked respiration, suggesting reduced ATP production in islet cells from male mice lacking *Insr* (Fig. 3j). Notably, RNA sequencing showed that male, but not female, βInsr^KO had significantly decreased expression of many key mitochondria-related genes (e.g., *Bckdk*, *Miga1*, *Park7*, *Me3*), including four components of the NADH:ubiquinone oxidoreductase complex (complex 1 of the electron transport chain; *Ndufb7*, *Ndufs6*, *Ndufb6*, *Ndufa2*), 2 components of ubiquinol-cytochrome c reductase complex (electron transport chain complex 3; *Uqcr11*, *Uqcrb*), and 2 components of ATP synthase (*Atp5g1*, *Atp5md*) (Fig. 2d, f). Male-specific transcriptomic consequences of *Insr* loss could account for the sex difference in mitochondrial function that would be expected to impact ATP-dependent membrane potential and Ca^2+

oscillations. Collectively, these experiments demonstrate that β-cells lacking *Insr* have increased electrical activity, so long as their mitochondrial ATP production is not impaired. Overall, our data support the concept that insulin normally has a negative feedback influence on excitability in the context of stimulatory glucose.

**Loss of β-cell Insr causes insulin hypersecretion in the context of stimulatory glucose.** Insulin secretion is driven by electrical excitability, so we next carefully examined the effects of partial and full β-cell *Insr* deletion on secretory function employing multiple orthogonal in vitro and in vivo assays. We used islet perifusion to examine the dynamics of insulin secretion ex vivo at rest (3 mM glucose) and in response to 20 mM glucose or 10 mM glucose, as well as direct depolarization with 30 mM KCl. Islets from female 16 week-old βInsr^KO and βInsr^HET mice secreted more insulin in response to 20 mM glucose and 30 mM KCl compared to islets from control mice (Fig. 4a). No significant differences were observed at low glucose (Fig. 4a). Consistent with our electrophysiology data, we did not observe differences in islets from males of the same age and on the same diet (Fig. 4b).

This potentiation of high glucose-stimulated insulin secretion ex vivo in the complete and partial *Insr* knockout β-cells, led us to examine how insulin levels were affected by glucose stimulation in vivo using the hyperglycemic clamp technique in awake mice. For this cohort of mice, there were no significant differences in body mass (control 20.5 ± 0.5 g *n* = 8 vs. βInsr^KO 20.8 ± 0.4 g *n* = 10), lean mass (control 17.5 ± 0.4 g vs. βInsr^KO 17.3 ± 0.2 g), or fat mass (control 1.8 ± 0.3 g vs. βInsr^KO 2.2 ± 0.2 g). Glucose infusion rates were adjusted in order to reach hyperglycemic levels (~19 mM) in βInsr^KO and wild type control mice. Interestingly, slightly higher glucose infusion rates were necessary in female βInsr^KO mice in comparison to control mice in order to reach similar hyperglycemic levels (Fig. 4c). In accordance with our ex vivo insulin secretion data, glucose-stimulated insulin secretion was higher in female, but not in male βInsr^KO mice compared with control mice (Fig. 4c–h). We further tested whether in vivo insulin secretion would be potentiated after a single bolus of glucose in mice with reduced β-cell *Insr*. At 11 weeks of age, plasma insulin response, relative to baseline, was elevated 30 min after i.p. injection of 2 g glucose/kg body mass in female (*p* = 0.06), but not in male, βInsr^KO mice compared to controls (Fig. 4i, j). In accordance with the electrophysiology, Ca^2+ oscillation, and islet perifusion data, we detected no statistical difference between female genotypes in fasting plasma insulin in vivo at multiple ages (Fig. S3). Together, these experiments suggest that β-cell Insr can play suppressive role in

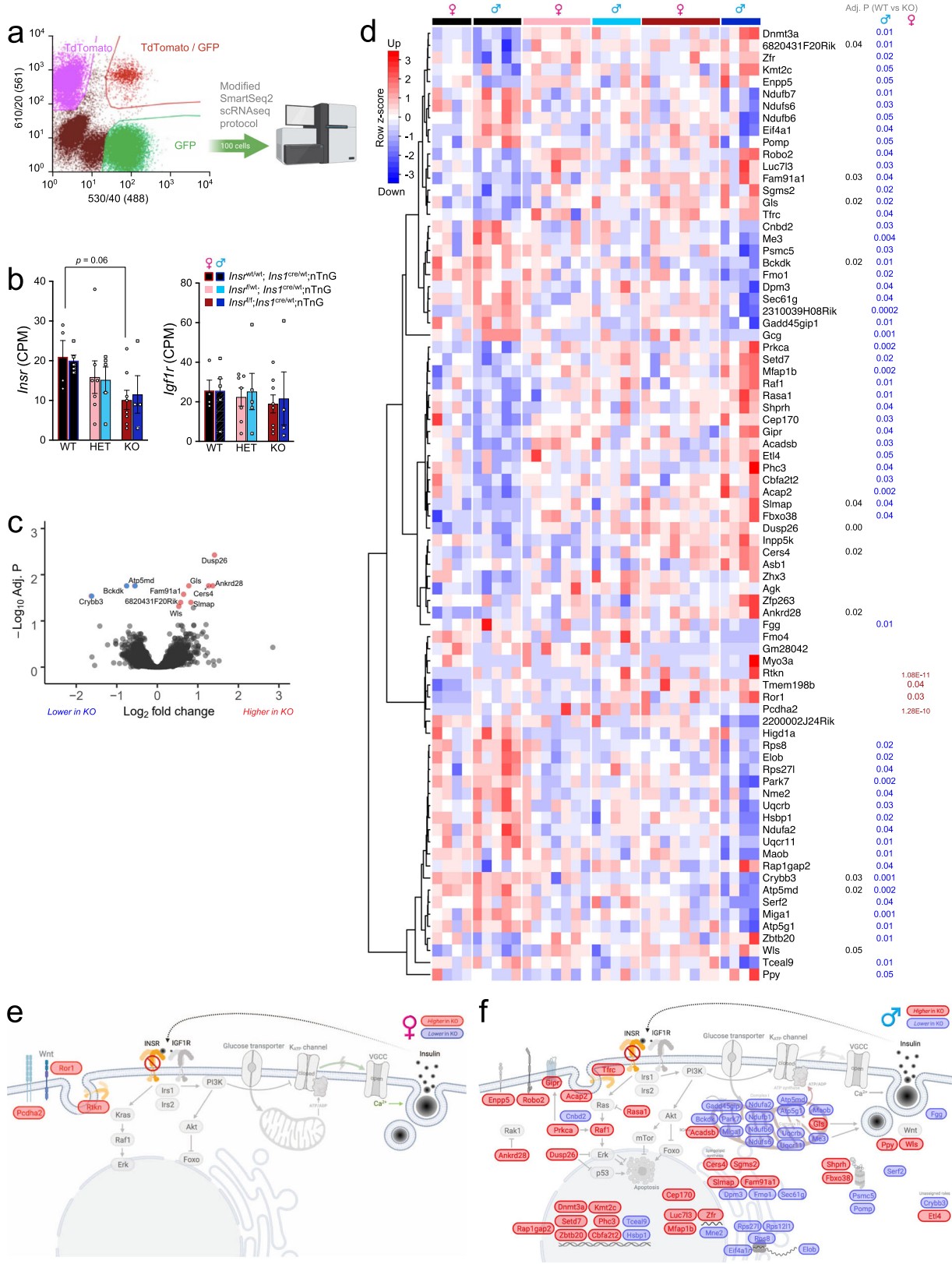

glucose-stimulated insulin secretion without much impact on basal insulin secretion.

### Effects of β-cell Insr loss on insulin production, storage, processing, and clearance.

Insulin and insulin signaling can modulate protein synthesis in many cell types and we have previously provided evidence that soluble cellular insulin protein transiently increases in human islet cell cultures treated with exogenous insulin[13]. To assess the quantitative contribution of Insr signaling to insulin production and mRNA translation rates we measured total islet insulin content after acid-ethanol extraction and examined total protein synthesis rate using S[35]- methoioine/

**Fig. 2 Transcriptomic analysis of purified *Insr* deficient β-cells. a** Workflow for RNA sequencing of FACS purified GFP-positive β-cells (100 per group) from 12-week old β*Insr*KO and littermate control mice. **b** *Insr* and *Igf1r* mRNA levels from 100 GFP positive β-cells from individual β*Insr*WT (female = black bar with red line, female = black bars with blue line), β*Insr*HET (female = light pink bars, male = light blue bars) and β*Insr*KO (female = dark red bars, male = dark blue bars) female (*n* = 4, 7, 8) and male (*n* = 5, 5, 4) mice. All samples were handled in a strictly blinded manner. Data were analysed by 2-way ANOVA with correction for multiple comparisons using Dunnett's method. Data are presented as mean values +/− SEM. Source data are provided as Source Data file. **c–f** Genes expressed higher or lower in β*Insr*KO vs. β*Insr*WT are indicated by red and blue, respectively. **c** Volcano plot and significantly differentially genes across both sexes (*n* = see **b**). **d** Heatmap of all genes that were differentially expressed between any group, clustered by similarity. *P* values are shown on the right for the β*Insr*WT to β*Insr*KO comparison for all, males, and females (*n* = see **b**). **e** Differentially expressed genes between female β*Insr*WT and β*Insr*KO cells overlaid on a diagram of their predicted functional roles and subcellular locations. **f** Differentially expressed genes between male β*Insr*WT and β*Insr*KO cells overlaid on a diagram of their predicted functional roles and subcellular locations.

cysteine pulse labeling in isolated islets. Total insulin content and protein synthesis in isolated islets were unaffected by *Insr* deletion under these basal glucose conditions (Fig. 4k–n).

To investigate the role of Insr signaling on β-cell stress and insulin clearance, we conducted analysis of plasma proinsulin to C-peptide ratios, and C-peptide to insulin ratios in the fasting state (4 h) across multiple ages in both male and female mice, and on multiple diets (Fig. S4). While many of these parameters changed with age, no statistical differences between genotypes were seen in mice fed either a low-fat diet (LFD) or a high fat diet (HFD). A trend towards lower insulin clearance was observed in LFD-fed female β*Insr*KO mice in comparison to wild type control mice at 7 weeks. Collectively, these experiments show that β-cell insulin receptor signaling has only a small, if any, effect on insulin processing and clearance at baseline glucose conditions.

**Beta-cell area and hyperglycemia-induced proliferation in mice lacking β-cell Insr.** Our RNA sequencing data revealed pathways downstream of *Insr* signaling that may affect β-cell proliferation capacity (e.g., *Raf1*)[16], specifically in male mice (Fig. 2f). Thus, we examined baseline β-cell area and proliferation reserve capacity. Islet architecture and β-cell-to-α-cell ratio were not obviously perturbed (Fig. 5a). We did not detect significant differences associated with genotype in β-cell area in either female or male mice, at 13, 42 weeks (LFD), or 54 weeks (HFD) of age (Fig. 5b–e). In comparison to control mice, HFD fed female β*Insr*KO mice had a tendency toward a smaller β-cell area at 54 weeks of age that was consistent with tendencies towards lower plasma insulin (Fig. 5d and S3c), proinsulin and C-peptide levels (Fig. S4c). These data suggest that Insr may help support age-dependent β-cell expansion under some conditions.

Prolonged hyperglycemia can stimulate β-cell proliferation in adult mouse β-cells[35], but whether this requires intact insulin receptor signaling remains controversial. To examine the role of *Insr*-mediated signaling on hyperglycemia-induced β-cell proliferation, we performed 4-day hyperglycemic infusions in β*Insr*KO and wild type control mice. In female mice, hyperglycemia (>10 mM; Fig. 5f) resulted in mildly elevated insulin secretion in β*Insr*KO relative to control mice for the initial 48 h, which was not sustained for the duration of the experiment (Fig. 5g, h), while glucagon levels declined similarly in both genotypes (Fig. 5i, j). There was no effect of *Insr* deletion on hyperglycemia-induced proliferation of either β-cells or α-cells in females (Fig. 5k, l). In male mice, 96 h of hyperglycemia resulted in sustained hyper-insulinemia in β*Insr*KO mice (Fig. 5m–o), with no differences in circulating glucagon (Fig. 5p, q). In male mice lacking β-cell *Insr*, this manipulation was associated with significantly more β-cell proliferation (Fig. 5r, s). The fact that we did not observe a suppression of glucose-induced proliferation of β-cells lacking *Insr* prompted us to determine the degree to which the, broadly defined, insulin signaling pathway was inhibited in our model. Indeed, glucose-induced Akt phosphorylation, shown by western blot of whole islet lysate, was statistically unaffected, and glucose-

induced Erk phosphorylation was only reduced ~50% in *Insr* knockout β-cells (Fig. 5t). It is likely that the *Igf1r* or another receptor tyrosine kinase (or a receptor tyrosine-kinase-independent mechanism) initiates components of intracellular post-receptor "insulin signaling" in the absence of *Insr*. Testing this hypothesis in the future will require truly β-cell specific double deletion of *Insr* and *Igf1r*, and/or additional receptors.

**Modeling contributions of peripheral and β-cell specific insulin sensitivity to glucose homeostasis.** The continuum between insulin sensitivity and resistance impacts multiple tissues, including pancreatic β-cells. We observed that β-cell-specific insulin resistance resulted in insulin hypersecretion in the context of unchanged β-cell mass. We next used mathematical modeling to generate quantitative predictions of the dependence of glucose tolerance on both β-cell and whole-body insulin resistance, both independently and synchronously. As described in the methods section, we modified the Topp model[36], adding insulin receptor-mediated negative feedback on insulin secretion, as indicated by our experimental data, with $S_β$ serving as the β-cell Insr-specific insulin sensitivity parameter (see the "Methods" section; for β*Insr*KO mice, $S_β = 0$). Peripheral insulin sensitivity is represented by $S_P$, ($S_I$ in the original Topp model). We used our hyperglycemic clamp data (Figs. 4c–e, f–h and S5a, b) to estimate $S_β$ in both female and male control mice and found $S_{β,female}$ to be significantly different from zero ($S_{β,female} = 3.4 ± 1.5$ nM$^{-1}$, $p = 10^{-25}$) and significantly different from $S_{β,male}$ ($p = 10^{-24}$). $S_{β,male}$ was not significantly different from zero ($S_{β,male} = -0.05 ± 1.0$ nM$^{-1}$, $p = 0.7$). In silico glucose tolerance tests found that decreased β-cell insulin-sensitivity ($S_β$) (similar to β*Insr*KO mice) corresponded with elevated peak and plateau insulin secretion (Fig. 6a). As expected, the computations predicted more rapid clearance of blood glucose (Fig. 6b). Analysis of areas under the curve for glucose and insulin resulting from in silico glucose tolerance tests while varying $S_P$ and $S_β$ indicated that β-cell insulin resistance should have a marked effect on insulin secretion and glucose tolerance, most dramatically in conditions of low peripheral insulin sensitivity (Fig. 6c, d). Next, we compared the predictions of this model with experimental results. We used the in silico AUC$_{Glucose}$ values as a function of both $S_P$ and $S_β$ combined with averaged experimental AUC$_{Glucose}$ values to estimate $S_P$ as a function of age for the low-fat diet conditions (Fig. 6f). We found that male values of $S_P$ for wildtype and mutant were indistinguishable from each other while females showed significant differences from each other and from the male values at all ages. As expected, HFD led to reduced peripheral insulin sensitivity (Fig. 6g). Collectively, these simulations show how β-cell insulin sensitivity and peripheral insulin sensitivity may combine to regulate glucose tolerance.

**Context-dependent improvement in glucose tolerance with reduced β-cell Insr signaling.** To test our theoretical model

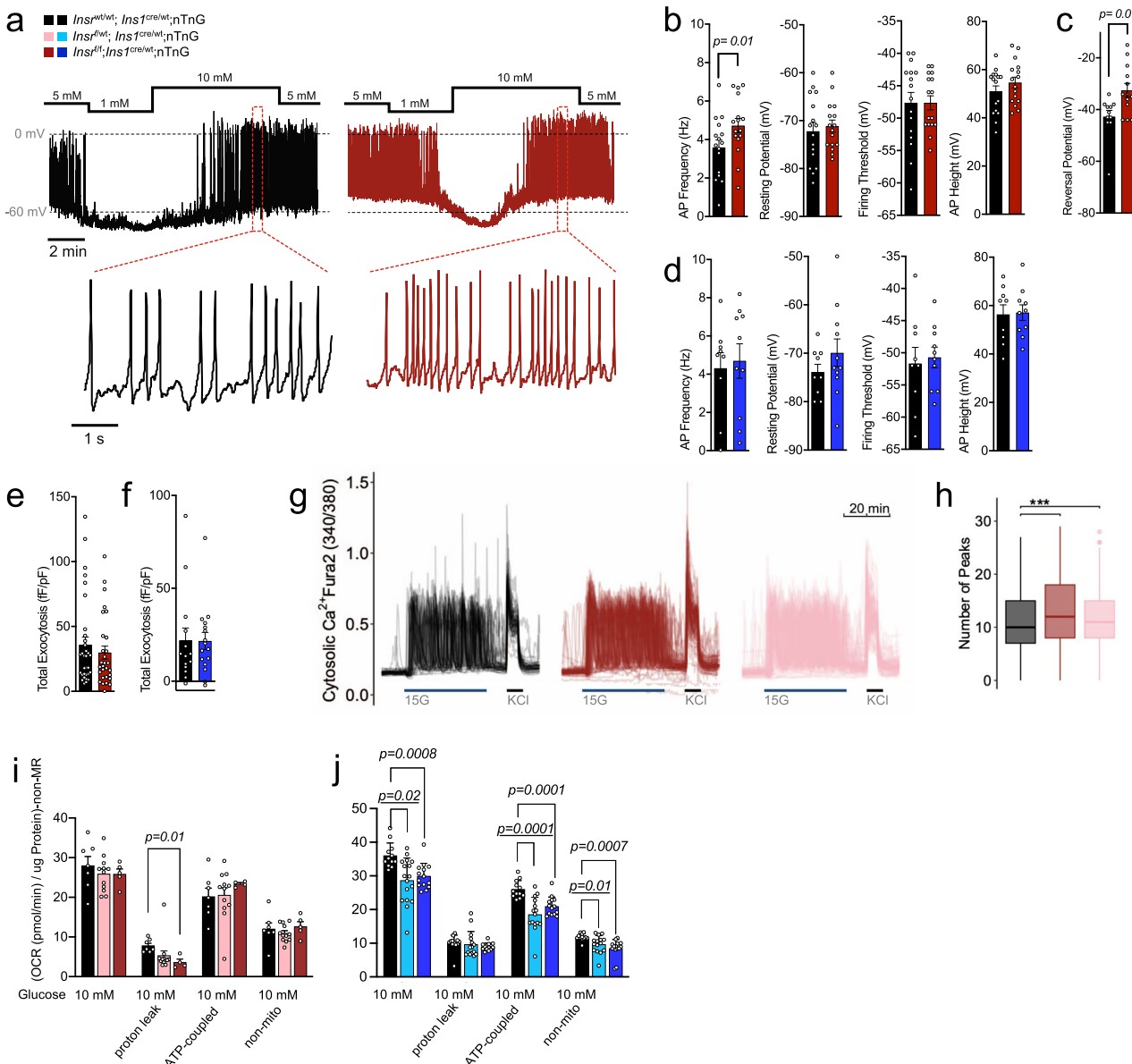

**Fig. 3 β-Cells lacking *Insr* have increased action potential and calcium oscillation frequencies. a** Representative traces of action potential firing in β-cells from 16 week-old chow-fed female mice (*n* = 16–17 cells, from three mice per group). Glucose changed as indicated. **b**, **c** Quantification of action potential (AP) properties (Hz, mV) and reversal potential (mV) during the 10 mM glucose phase in patch clamped dispersed female β-cells (*n* = 16–17 cells, from three mice per group). Unpaired two-sided student's *t*-test. **d** Quantification of electrophysiological properties in islets from male mice (*n* = 9–10 cells, from three mice per group). Unpaired two-sided student's *t*-test. **e**, **f** β-Cell exocytosis measured as increased membrane capacitance normalized to initial cell size (fF/pF) over a series of ten 500 ms membrane depolarizations from −70 to 0 mV (*n* = control 32 cells, 29 β*Insr*KO cells, from three pairs of mice). Unpaired two-sided student's *t*-test. **g** Ca²⁺ dynamics measured in dispersed islet cells (Fura-2 340/380 ratio) treated as indicated from a baseline of 3 mM glucose) (*n* = 3523 cells, 3–5 image areas per experiment, 1–2 independent islet cultures from six LFD-fed female 12-week old mice, two mice from each group). **h** Quantification of glucose induced Ca²⁺ oscillation number (*n* = see **g**). ANOVA with correction for multiple comparisons using Tukey's method. Additional quantification of these traces can be found in Fig. S2A. Boxplot; Minima: minimum outliers; Maxima: maximum outliers; Center: median; Bounds of box: first to third quartile; Whiskers: the upper whisker extends from the hinge to the largest value no further than 1.5 * IQR from the hinge (where IQR is the inter-quartile range, or distance between the first and third quartiles). The lower whisker extends from the hinge to the smallest value at most 1.5 * IQR of the hinge. **i**, **j** Oxygen consumption rate (OCR) data ((pmol/min) normalized to ug protein with non metabolic rate (non-MR) subtracted) of independent dispersed islet cultures from 16 week-old chow-fed control, β*Insr*HET, and β*Insr*KO mice (three male and one female mice from each group). Data were analyzed with a fitted mixed-effects model with correction for multiple comparisons using Dunnett's method. β*Insr*WT (black bars), β*Insr*HET (female = light pink bars, male = light blue bars) and β*Insr*KO (female = dark red bars, male = dark blue bars). All samples were handled in a strictly blinded manner. All data are presented as mean values +/− SEM. Source data are provided as Source Data file.

experimentally, we examined glucose tolerance in female and male β*Insr*KO, β*Insr*HET, and control littermates at multiple ages between 4 and 52 weeks in the context of two diets. Significant improvements in glucose tolerance were observed in female mice with reduced β-cell Insr signaling at multiple ages, and in young males (Fig. 7). Consistent with our mathematical modeling that suggested a diminished contribution of β-cell insulin resistance to glucose homeostasis in the context of greater whole-body insulin

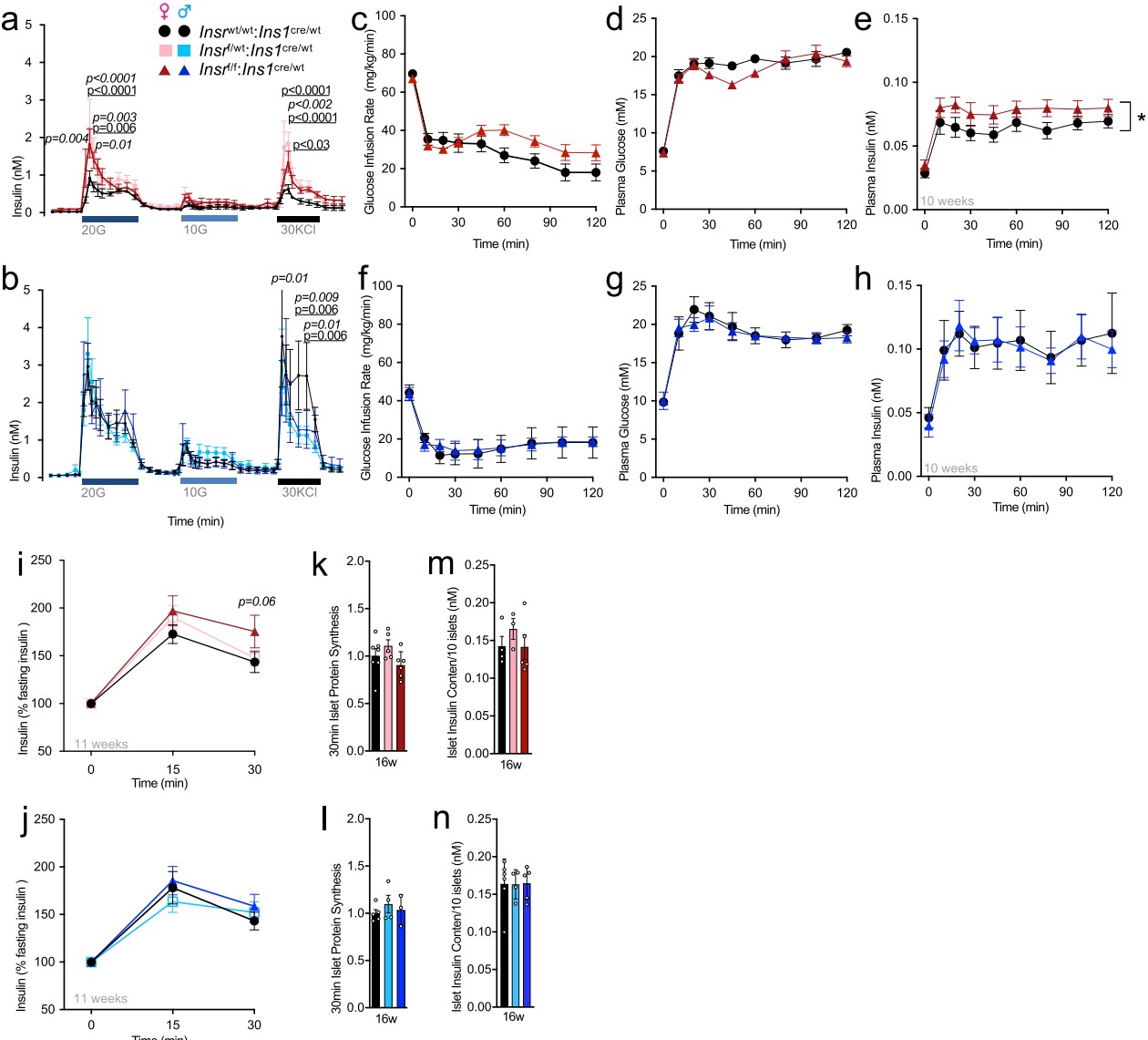

**Fig. 4 Insr knockout increases glucose-stimulated insulin secretion in vitro and in vivo. a, b** Insulin levels (nM) from perifused islets isolated from 16 week old chow-fed control, βInsr[HET], βInsr[KO] female (two parallel runs per $n = 3, 3, 5$), and male (2 parallel runs per $n = 6, 5, 5$) mice were exposed to 20 mM glucose (20 G), 10 mM glucose (10 G), and 30 mM KCL (KCL) from a baseline of 3 mM glucose. Data analyzed using repeated measures mixed effects model. *p-values are italicized when βInsr[KO] was compared to controls, p-values are underlined when βInsr[HET] was compared to controls. Quantification of 1st phase and 2nd phase during 20 mM glucose stimulation, total response during 10 mM glucose stimulation, and during 30 mM KCL stimulation. AUC's were analyzed with 1-way ANOVA analysis. **c–h** Glucose infusion rates (mg/kg/min), plasma glucose levels (mM), and plasma insulin (nM) levels during 2-h hyperglycemic clamps in awake LFD-fed, 10-week old control and βInsr[KO] female ($n = 14, 15$) and male ($n = 4, 4$) mice. Data were analyzed using repeated measures mixed-effects model. **i, j** Insulin levels (% basal insulin) following a single glucose injection (2 g glucose/kg body mass, i.p) of 11 week old LFD-fed control, βInsr[HET], βInsr[KO] female ($n = 33, 34, 23$) and male ($n = 22, 29, 17$) mice. Data were analyzed using repeated measures mixed-effects model. **k, l** Islet protein synthesis measured (fold change) for 30 minutes by S[35] labeling in islets isolated from individual control, βInsr[HET], βInsr[KO] female ($n = 7, 5, 6$) and male ($n = 5, 4, 3$) mice. Data were analyzed by one-way ANOVA. **m, n** Insulin content (nM) of ten islets isolated from control, βInsr[HET], βInsr[KO] female ($n = 4, 3, 5$) and male ($n = 5, 4, 5$) mice. Data were analyzed by one-way ANOVA. βInsr[WT] (black bars, lines, circles), βInsr[HET] (female = light pink bars, lines, squares; male = light blue bars, lines, squares) and βInsr[KO] (female = dark red bars, lines, triangles; male = dark blue bars, lines, triangles). All animals/samples were handled in a strictly blinded manner. All data are presented as mean values +/− SEM. Source data are provided as Source Data file.

resistance, we did not observe significant effects of genotype in older male mice, or mice of either sex fed an insulin-resistance-inducing HFD, in contrast to models with both β-cell and brain *Insr* knockout[20–22]. Thus, *Insr* deletion specifically in β-cells had less effect on glucose tolerance in mice with already impaired pan-tissue insulin resistance, which we and others have shown

increases with age and is more pronounced in male mice (Figs. 6f and S6).

The multiple analyses conducted on both sexes, at different ages, and on one of two diets can be appropriately analyzed using Bayesian methods. Similar to the frequentist statistical analysis, we observed evidence for improvements in glucose tolerance in female

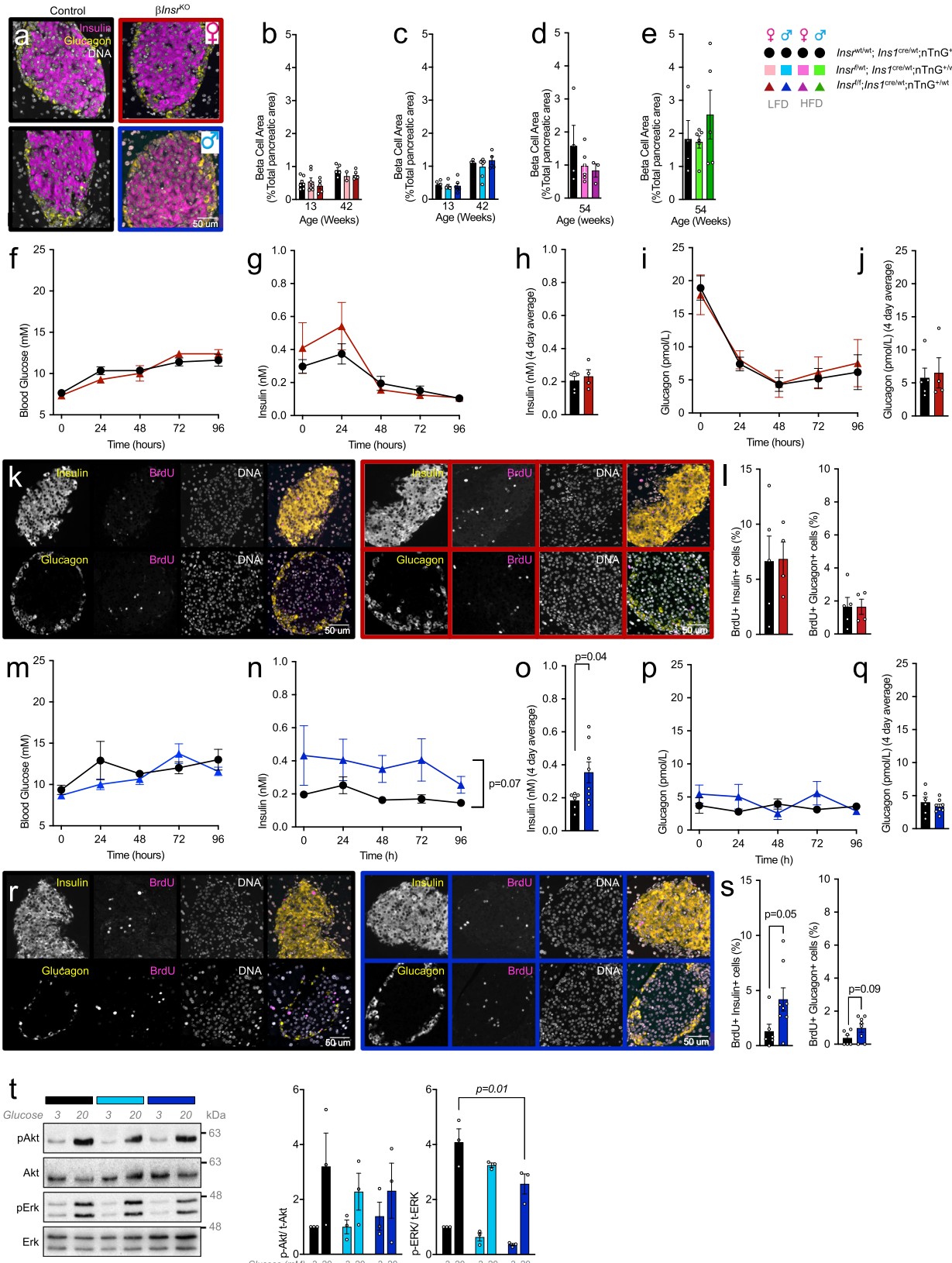

mice with reduced Insr signaling at 9 and 39 weeks of age, and in males at 4 and 39 weeks of age (Fig. 8). In the context of high fat diet-induced insulin resistance, we only observed evidence for improved glucose tolerance in older female mice without β cell Insr signaling. Generally, this statistical analysis validates and extends the more commonly used statistical methods above.

**Acutely improved glucose tolerance with inducible β-cell-specific Insr loss.** The $Ins1^{cre}$ allele results in pre-natal gene deletion[30]. To assess the effects of $Insr$ deletion in adult β-cells, and to determine the role of $Insr$ on a different genetic background and under different housing conditions, we also phenotyped multiple cohorts of male mice in which the $Insr^{f/f}$ allele was

**Fig. 5 Islet cell proliferation and relative area in mice lacking β-cell *Insr*. a** Representative images showing islet architecture via staining for insulin (pink), glucagon (yellow) and DNA (white) in control (black frame) and βInsr$^{KO}$ mice (female = red frame; male = blue frame). Similar results were observed in independent experiments (female $n = 5$ control, four βInsr$^{KO}$ and male $n = 6$ control, eight βInsr$^{KO}$ mice). **b–e** β-cell area shown as a percentage of total pancreatic area. βInsr$^{WT}$ (black bars), βInsr$^{HET}$ (LFD-fed female = light pink bars; LFD-fed male = light blue bars; HFD-fed female = bright pink bars; HFD-male = bright green bars) and βInsr$^{KO}$ (LFD-fed female = dark red bars; LFD-fed male = dark blue bars; HFD-fed female = purple bars; HFD-male = dark green bars). Data were analyzed by one-way ANOVA analyses. **f–s** 4-day in vivo glucose infusion in LFD-fed 10-week-old female control ($n = 5$, black lines, solid circles), βInsr$^{KO}$ ($n = 4$, dark red lines, solid triangles) and male control ($n = 6$, black lines, solid circles), βInsr$^{KO}$ ($n = 8$, dark blue lines, solid triangles) mice. **f, m** Tail blood glucose (mM) and **g, n** insulin (nM) data were analyzed using repeated measures mixed effects models. **h, o** Four days average plasma insulin (nM) data were analyzed using an two-sided unpaired student's *t*-test. **i, p** Plasma glucagon (pmol/L) (tail blood) data were analyzed using repeated measures mixed-effects model. **j, q** Four days average plasma glucagon (pmol/L) data were analyzed by an two-sided unpaired student's *t*-test. **k, r** Representative single channel (white) and merge images showing islets stained for insulin (yellow, upper panel), glucagon (yellow, lower panel), BrdU (pink) and DNA (white) from LFD-fed female and male controls and female βInsr$^{KO}$ and male βInsr$^{KO}$ mice following four day glucose infusion. Similar results were observed in independent experiments (female control $n = 5$, female βInsr$^{KO}$ $n = 4$, male control $n = 6$, male βInsr$^{KO}$ $n = 8$). **l, s** Quantification of BrdU+ insulin+ cells (% all insulin+ cells) and BrdU+ glucagon+ cells (% all glucagon+ cells). Data were analyzed by two-sided unpaired Student's *t*-tests. **t** Representative western blot image and quantification of islet lysate from LFD-fed 12-week old male Insr$^{wt/wt}$;Ins1$^{cre/wt}$;nTnG (black bar, $n = 3$), Insr$^{f/wt}$;Ins1$^{cre/wt}$;nTnG (light blue bar, $n = 3$), Insr$^{f/f}$;Ins1$^{cre/wt}$;nTnG (dark blue bar, $n = 3$) mice treated with 3 or 20 mM glucose. Data were analyzed by one-way ANOVA. All animals/samples were handled in a strictly blinded. All data are presented as mean values ± SEM. Source data are provided as Source Data file.

recombined by the *Ins1* promoter-driven CreERT transgenic allele (commonly known as MIP-Cre) after injection with tamoxifen. In agreement with our observations in mice with constitutive loss of β-cell *Insr*, we found that glucose tolerance was significantly improved 4 weeks after β-cell-specific *Insr* deletion in male mice (Fig. 9). These differences were not maintained as the mice aged and became more insulin resistant. In these mice, there were no significant differences observed in fasting glucose (control $4.8 ± 0.3$ mM $n = 7$ vs. βInsr$^{KO}$ $4.4 ± 0.2$ mM $n = 10$), β-cell mass (control $1.4 ± 0.3\%$ $n = 3$ vs. βInsr$^{KO}$ $2.1 ± 0.2\%$ $n = 5$), or body mass (control $26.1 ± 1.1$ g $n = 7$ vs. βInsr$^{KO}$ $23.1 ± 0.7$ g $n = 17$). Collectively, these observations using an independent model and independent housing conditions lend support to our conclusion that the initial consequence of β-cell specific *Insr* deletion is improved glucose tolerance. This experiment also suggests that the roles of Insr in β-cell function are not formally sex specific, just sex biased depending on the degree of peripheral insulin resistance.

**Peripheral effects of β-cell specific Insr loss**. We and others have shown that even modest differences in hyperinsulinemia can have profound consequences for insulin sensitivity, adiposity, fatty liver, longevity and cancer[14,37,38]. Thus, we asked how the context-dependent glucose-stimulated insulin hyper-secretion induced by targeted β-cell specific insulin resistance may affect insulin sensitivity, adiposity, and body mass over time. Insulin sensitivity was assessed at multiple ages in the same mice. Interestingly, insulin sensitivity was significantly improved in 10-week-old female βInsr$^{HET}$ mice compared to littermate controls without *Insr* deletion (Fig. S6a). On a high fat diet, male βInsr$^{KO}$ and βInsr$^{HET}$ mice had significantly improved insulin sensitivity compared to controls at 22 weeks of age (Fig. S6d). Longitudinal tracking of 4-h fasting blood glucose identified relative hypoglycemia in young LFD-fed female βInsr$^{KO}$ and βInsr$^{HET}$ mice, older LFD-fed male βInsr$^{KO}$ and βInsr$^{HET}$ mice, and across the tested ages in HDF-fed female mice (Fig. 10a–d). Longitudinal tracking of body weight revealed that female mice with reduced β-cell Insr consistently weighed more than controls when fed a HFD (Fig. 10e–h), consistent with the known role of hyperinsulinemia in diet-induced obesity[14,39]. We also examined the mass of several tissues at 13 weeks of age. Interestingly, liver mass was lower in both female and male mice lacking β-cell *Insr* (Fig. 10i, j). Pilot experiments suggested that liver Insr protein abundance may have been reduced in mice with partially or completely reduced β-cell Insr, in the context of the LFD but not the HFD (Fig. 10k).

These data are consistent with the concept that modest hyperinsulinemia can drive down Insr levels[40] and the idea that insulin is a trophic signal for liver. Together, these data demonstrate that specifically preventing autocrine insulin feedback can have systemic effects on insulin sensitivity, body mass, and the size of some tissues. Beta-cell specific insulin resistance resulting in modest hyperinsulinemia after repeated glucose stimulation may affect the eventual susceptibility to type 2 diabetes.

## Discussion

The goal of the present study was to establish the role of β-cell Insr on glucose homeostasis using specific genetic loss-of-function tools. We found that in vivo *Insr* gene deletion potentiated glucose-stimulated insulin secretion by increasing action potential firing and Ca$^{2+}$ oscillation frequency, leading to improved glucose tolerance in insulin sensitive animals. Our data therefore suggest a model in which insulin inhibits its own secretion in a context-dependent manner and that this local negative feedback loop has physiological consequences for glucose tolerance.

Autocrine signaling in endocrine cells is generally a negative feedback[41], with a few exceptions in specific conditions[42]. Given the abundance of Insr protein in islets and the physiological modulation of both insulin and Insr signaling in health and disease, autocrine and paracrine insulin signaling have been topics of interest and controversy for decades[6,43]. While some have questioned whether local insulin levels are sufficient for signaling within the islet, mathematical modeling of insulin hexamer dissolution estimated that monomeric insulin within islets is in the picomolar range[44], similar to the dose that maximally activates β-cell insulin signaling pathways[16,45]. Consistent with a narrow range of responsiveness, our results also show that the contribution of autocrine insulin feedback to glucose homeostasis depends on whether mice are on a diet or at an age where insulin resistance is high, and potentially saturated in β-cells. Background genetics, diet, housing conditions, microbiome, or glucose concentrations could contribute to differences in observed phenotypes between our β-cell specific *Insr* knockout models and the frank diabetes reported for transgenic models that use fragments of the *Ins2* or *Pdx1* promoters to drive Cre-mediated β-cell *Insr* deletion[20–22]. However, we believe that the discrepancy is more likely due to depletion of *Insr* in key neuronal populations since both of *Ins2* and *Pdx1* are expressed in brain regions that influence glucose homeostasis[23,46]. For example, *Insr* deletion in the brain with Nestin Cre causes insulin resistance[47]

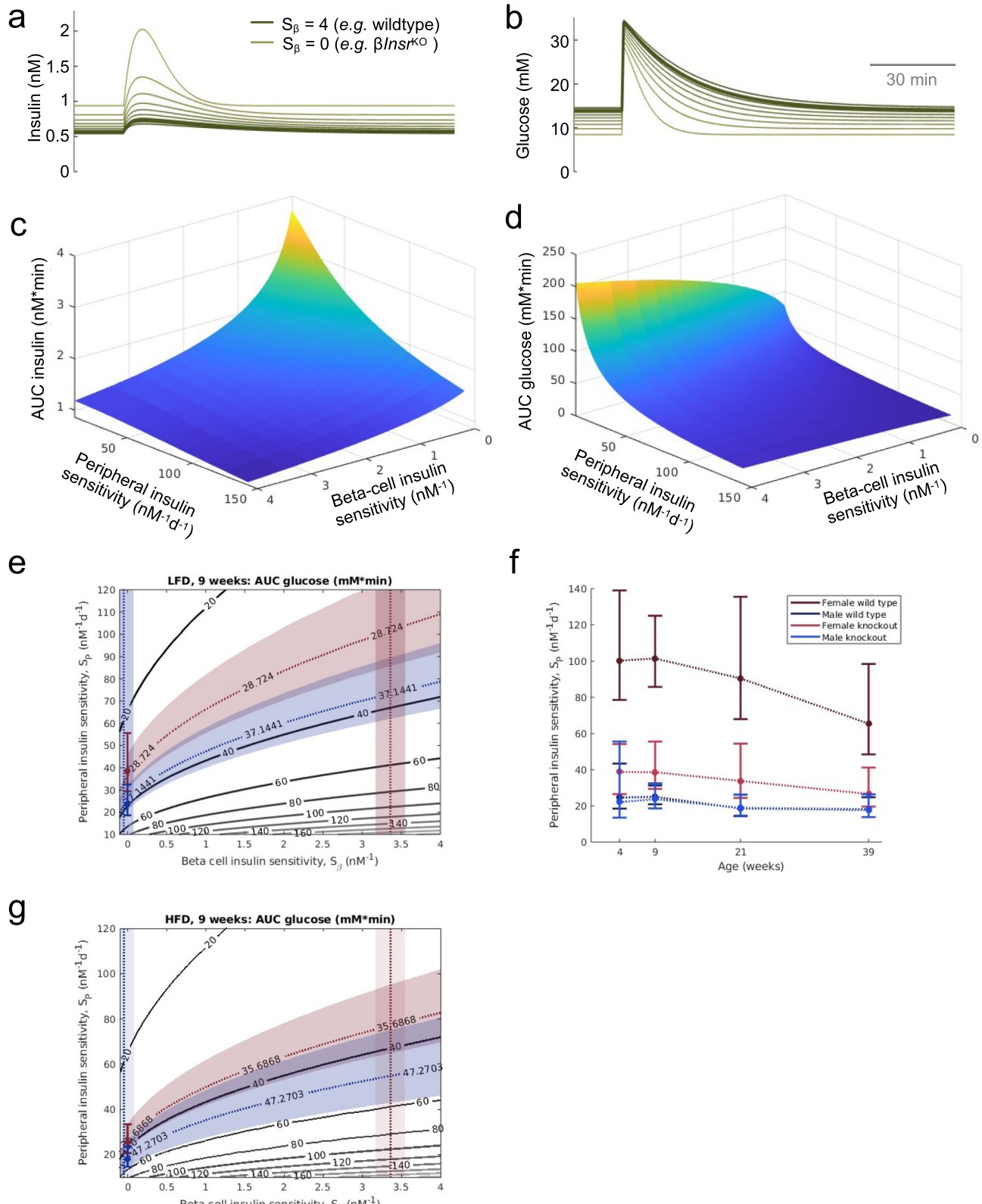

**Fig. 6 Mathematical modeling of insulin secretion, glucose tolerance and insulin sensitivity.** The experimental data are the same as Figs. 4 and S6. **a**, **b** Simulations of the effects of reduced β-cell insulin sensitivity on glucose stimulated insulin release and glucose tolerance. **c**, **d** Relationship between the contributions of peripheral insulin sensitivity and β-cell insulin sensitivity to the glucose AUC and insulin AUC in the in silico glucose tolerance tests. **e** Illustration of our method for estimating $S_P$ using experimental glucose AUC and $S_\beta$ (as estimated above) for the week 9 male (blue) and female (red) LFD mice. Experimental glucose AUC values (mean ± SEM) give bands of possible $S_\beta$ and $S_P$ values using the glucose AUC surface in **d**. The intersection of these bands with the band for our $S_\beta$ estimates give a range of values for the $S_P$ estimates. **f** The resulting $S_P$ estimates for weeks 4, 9, 21, 39 summarized in a single plot. **g** $S_P$ estimate for HFD mice at 9 weeks. All data are presented as mean values ± SEM. Source data are provided as Source Data file.

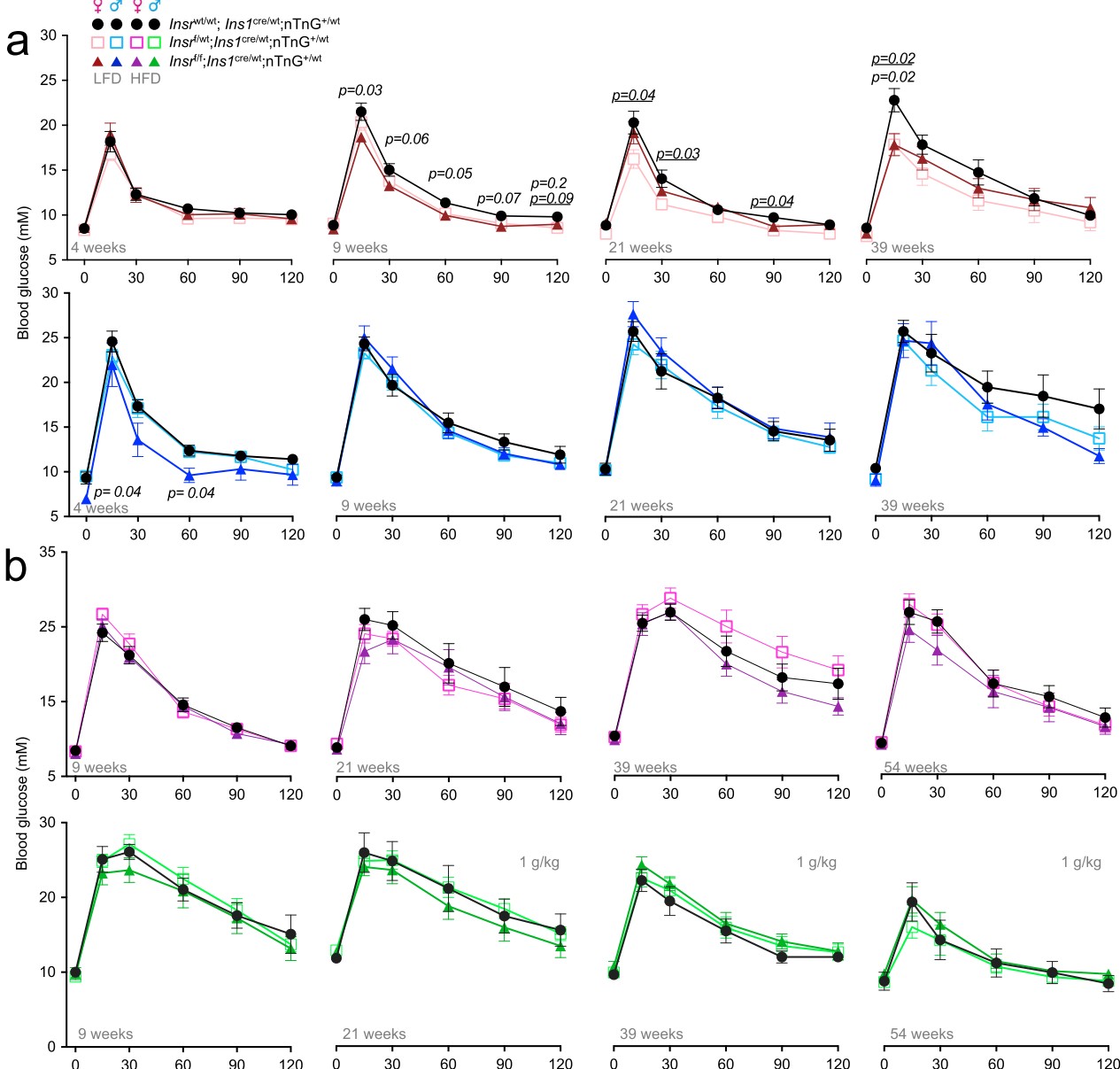

**Fig. 7 Glucose tolerance at multiple ages in β-cell specific *Insr* knockout mice fed 2 different diets. a** Tail blood glucose levels (mM) from glucose tolerance tests of LFD-fed females (control (black lines, solid circles), β*Insr*HET (light pink, open squares), β*Insr*KO, (dark red lines, solid triangles)); ($n_{4week}$ = 16, 16, 14; $n_{9week}$ = 23, 25, 17; $n_{21week}$ = 12, 15, 12; $n_{39week}$ = 8, 12, 10) and males (control (black lines, solid circles), β*Insr*HET (light blue, open squares), β*Insr*KO (male = dark blue, solid triangles); ($n_{4week}$ = 10, 18, 5; $n_{9week}$ = 17, 30, 11; $n_{21week}$ = 8, 14, 8; $n_{39week}$ = 9, 7, 13). **b** Tail blood glucose levels (mM) from glucose tolerance tests of HFD-fed female (control (black lines, solid circles), β*Insr*HET (bright pink, open squares), β*Insr*KO (purple lines, solid triangles)); ($n_{9week}$ = 17, 17, 14; $n_{21week}$ = 12, 14, 12 $n_{39week}$ = 14, 16, 10; $n_{54week}$ = 10, 13, 11) and male (control (black lines, solid circles), β*Insr*HET (bright green, open squares), β*Insr*KO (dark green lines, solid triangles)); ($n_{9week}$ = 7, 16, 9; $n_{21week}$ = 7, 16, 8; $n_{39week}$ = 8, 14, 8; $n_{54week}$ = 6, 11, 7) mice. All mice received a glucose bolus of 2 g glucose/kg body mass (*i.p*) except older HFD-fed males, which received only 1 g glucose/kg body mass (*i.p*). *p*-values are italicized when β*Insr*KO was compared to controls, *p*-values are italicized and underlined when β*Insr*HET was compared to controls. Data were analysed by a fitted mixed-effects model with correction for multiple comparisons using Dunnett's method. All animals/samples were handled in a strictly blinded manner. All data are presented as mean values ± SEM. Source data are provided as Source Data file.

and impairs the sympathoadrenal response to hypoglycemia[48], while *Insr* knockout using AgRP Cre results in abnormal suppression of hepatic glucose production[49]. Our β*Insr*KO mouse line is the most tissue specific model used to date for the study of autocrine insulin signaling and β-cell insulin resistance.

Using genetic and genomic tools, our work complements previous studies in humans and animal models. Ex vivo studies of perfused canine pancreata found an inhibitory autocrine effect of insulin[50]. Similarly, exogenous insulin perfusion of canine

pancreas in situ was shown to lead to reduced endogenous insulin production[51]. In vivo insulin infusion rapidly suppressed C-peptide levels in healthy men, but not those with obesity and presumably global insulin resistance[8]. In isolated human islets, perifusion studies showed that treatment with physiological doses of insulin had no effect on C-peptide release[11], while static incubation experiments found only moderate potentiation of glucose-stimulated insulin secretion with super-physiological levels of insulin[43]. Our conclusions are in line with the recent

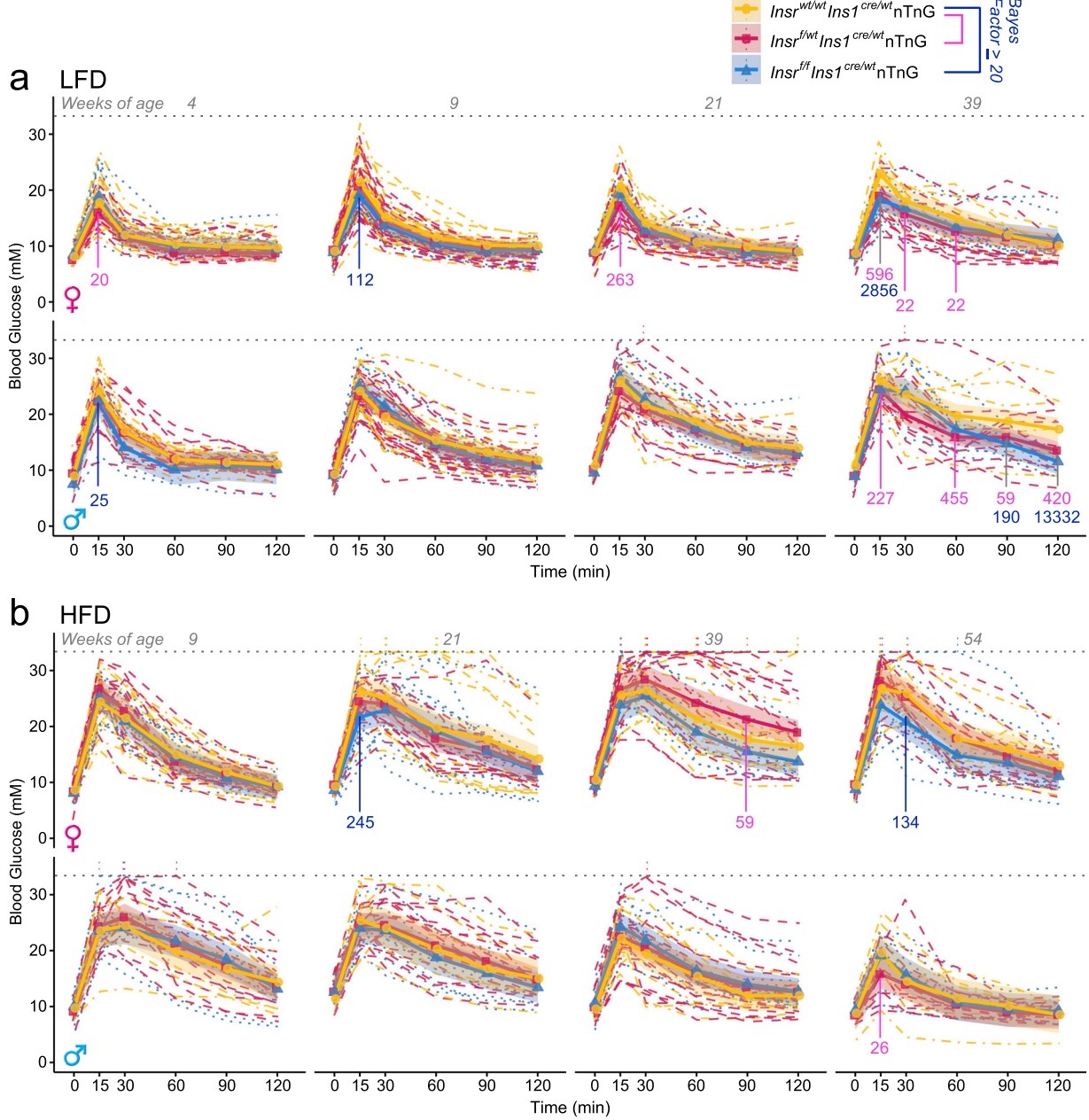

**Fig. 8 Bayesian regression modeling of glucose tolerance at multiple ages in β-cell specific *Insr* knockout mice fed two different diets.** Modelled from the same data as Fig. 7. **a** Tail blood glucose levels (mM) from glucose tolerance tests of control (yellow), βInsr^HET (red), βInsr^KO (blue) LFD-fed female as well as **b** HFD-fed female and male mice (*n* = see Fig. 7). All mice received a glucose bolus of 2 g glucose/kg body mass (*i.p*) except older HFD-fed males, which received only 1 g glucose/kg body mass (*i.p*). Data were analysed using Bayesian multilevel regression modeling. Bayes factors greater to or equal to 20 for individual time points are shown in pink for comparisons between control and βInsr^HET, and blue for comparisons between control and βInsr^KO. Additional Bayes factors can be found in Supplemental Table S3. Dashed lines indicate individual mice, solid lines and points indicate estimates, and shading indicates 95% credible intervals. Dotted line at 33.3 mM indicates the upper limit of detection of the glucometer and vertical dotted lines rising above this line indicate where measurements were above the limit of detection for an individual mouse. Source data are provided as Source Data file.

work of Paschen et al. reporting that a high fat, high sucrose diet induced tissue-selective insulin resistance in β-cells, as well as profound hyperinsulinemia and β-cell hyper-excitability[5]. Our mechanistic finding that *Insr* knockout β-cells have increased action potential firing frequency is consistent with previous observations that insulin directly increased $K_{ATP}$ currents via PI3-kinase signaling[33] and that PI3-kinase inhibition with wortmannin potentiates glucose stimulated insulin secretion in normal, but not T2D, human islets[52]. In the islets from 16-week-old male *Insr* knockout mice, it seems likely that the significant reduction

in ATP-production and differences in gene expression undercut this effect. Collectively, our studies further illustrate the molecular mechanisms of negative autocrine feedback in β-cells during high glucose stimulation.

Beyond the potentiation of glucose-stimulated insulin secretion, transcriptomic analysis of fully and partially Insr-deficient β-cells revealed a re-wiring of signaling pathways that could change how these cells respond to stresses (Fig. S1). For example, increased expression of the dual-specificity phosphatase *Dusp26* in βInsr^KO cells is expected to modulate Erk signaling

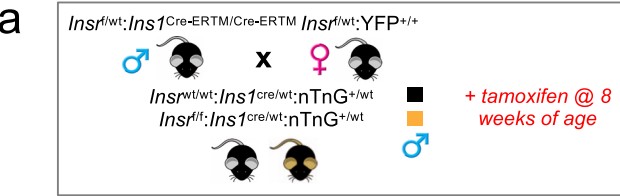

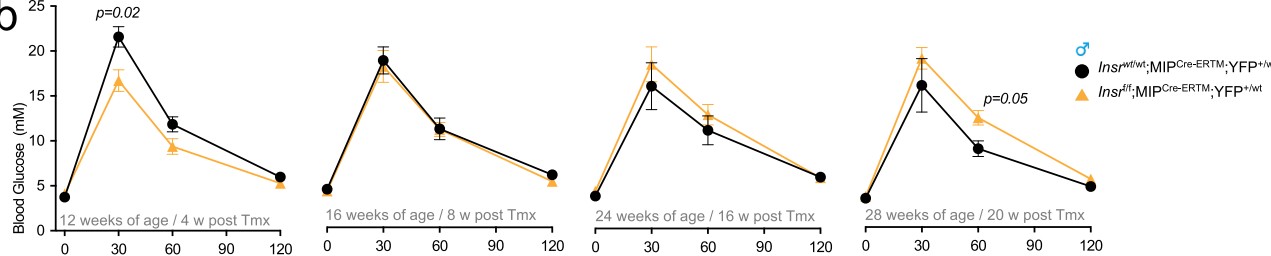

**Fig. 9 Tail blood glucose levels (mM) from glucose tolerance tests in inducible β-cell specific *Insr* knockout mice. a** Breeding and experimental design for inducible β-cell specific Insr knockout mice. **b** Glucose tolerance of Chow-fed *Insr*^wt/wt^;MIP^Cre-ERTM^;YFP^+/wt^ (black lines, solid circles, *n* = 7)) and *Insr*^f/f^;MIP^Cre-ERTM^;YFP^+/wt^ (yellow lines, solid triangles, *n* = 17) mice were examined at 4, 8, 16, and 20 weeks after tamoxifen injection at 8 weeks of age (3 × 200 mg/kg tamoxifen over a 1-week period). Data were analysed using repeated measures mixed-effects models. Data were analysed by a fitted mixed-effects model with correction for multiple comparisons using Dunnett's method. All data are presented as mean values ± SEM. Source data are provided as Source Data file.

downstream of Insr. Dusp26 has been implicated as a negative regulator of β-cell identity and survival[53]. Increased *Ankrd28*, a regulatory subunit of protein phosphatase 6[54], is expected to modulate inflammatory signaling. Increased ceramide synthase 4 (*Cers4*) would affect sphingolipid signaling and has shown to mediate glucolipotoxicity in INS-1 β-cells[55]. Increased mitochondrial glutaminase (*Gls*) would affect local glutamate synthesis, potentially affecting insulin secretion[56]. Increased *Wls* would support Wnt protein secretion. On the other hand, decreased branched-chain alpha-ketoacid dehydrogenase (*Bckdk*) would be expected to lower cellular levels of branched-chain amino acids. The consequences of decreased *Crybb3* are not clear in this context.

Our transcriptomic analysis pointed to robust sex dependent consequences of β-cell specific *Insr* deletion. Specifically in female βInsr^KO^ cells, we found increased expression of *Ror1*, a receptor tyrosine kinase implicated as a scaffold protein for caveolae-dependent endocytosis[57], which mediates *Insr* internalization and survival in β cells[31], an inhibitory scaffold of Rho GTPases (*Rtkn*), and the calcium-dependent adhesion protein (*Pcdha2*). The transcriptomic changes in male βInsr^KO^ cells were broad-based and included down-regulation of key genes controlling mitochondrial metabolism. There was also decreased *Hsbp1*, which links insulin signaling to longevity in *C elegans*[58]. The transferrin receptor (*Tfrc*) has been linked to insulin sensitivity and type 2 diabetes[59–61]. Male βInsr^KO^ cells also had increased *Rasa1*, which is a suppressor of Ras. Male βInsr^KO^ cells exhibited increased expression of signaling genes with known roles in β-cell function. Using MIN6 cells and primary β-cells from male mice, we previously showed that *Raf1* mediates many of insulin's effects on β-cell survival, proliferation and β-cell function[16,45,62,63]. Protein kinase C alpha (*Prkca*) is a direct upstream regulator of *Raf1* and may also stimulate the β-cell expression of incretin receptors such as *Gipr*[64]. *Robo2* plays important roles in β-cell survival[65] and islet architecture[66]. Male βInsr^KO^ cells also had up-regulation of pancreatic polypeptide (*Ppy*), but whether this reflects a change in β-cell differentiation status or the purity of sorted cells remains unclear. Additional studies, beyond the scope of this

investigation, will be required to assess the contribution of these differentially expressed genes towards the phenotype of female and male βInsr^KO^ mice.

The conditions under which these gene expression changes could manifest in long-term effects on β-cell proliferation or increased β-cell survival remain incompletely explored. Otani et al reported modestly reduced β-cell mass in non-diabetic Ins2-Cre transgenic Insr knockouts, which was exacerbated by diabetes, but they did not employ Cre controls[22]. Okada et al. reported impaired compensatory β-cell proliferation in the context of high fat diet or liver *Insr* knockout in the same experimental groups[21]. These previous studies comparing β-cell knockout mice of Insr versus Igf1r, suggested a more important role for the former in β-cell proliferation and survival[21]. *Insr* over-expression experiments also support the idea that β-cells are key insulin targets[67]. In the present study, we were unable to identify conditions which would result in statistically significant differences in β-cell mass in mice with Insr deficiency, but variability was high and relatively few animals were studied at older ages. We have previously shown that the increase in β-cell mass resulting from high fat diet requires local insulin production and is independent of hyperglycemia[14], consistent with the known direct anti-apoptotic and *Raf1*-dependent mitogenic effects of insulin in vitro[15,16,68]. These findings can be reconciled by proposing that high insulin concentrations within the islet are sufficient to activate remaining Igf1 receptors linked to *Raf1*-biased post-receptor signaling. We observed that sustained hyperglycemia and hyperinsulinemia over 4 days was associated with increased proliferation, but whether these are effects are mediated through Igf1r function compensation will require double receptor gene knockout experiments. The recent identification of a negative modulator of both Insr/Igf1r action in β-cells with profound effects on glucose homeostasis, supports the concept that double Insr/Igf1r inhibition inhibits β-cell proliferation[69]. It is also possible that hyperglycemia itself is a major driver of β-cell proliferation under these conditions, through Irs2-Creb signaling that may bypass Insr/Igf1r[18]. It has also been demonstrated that ~80% of the gene expression changes

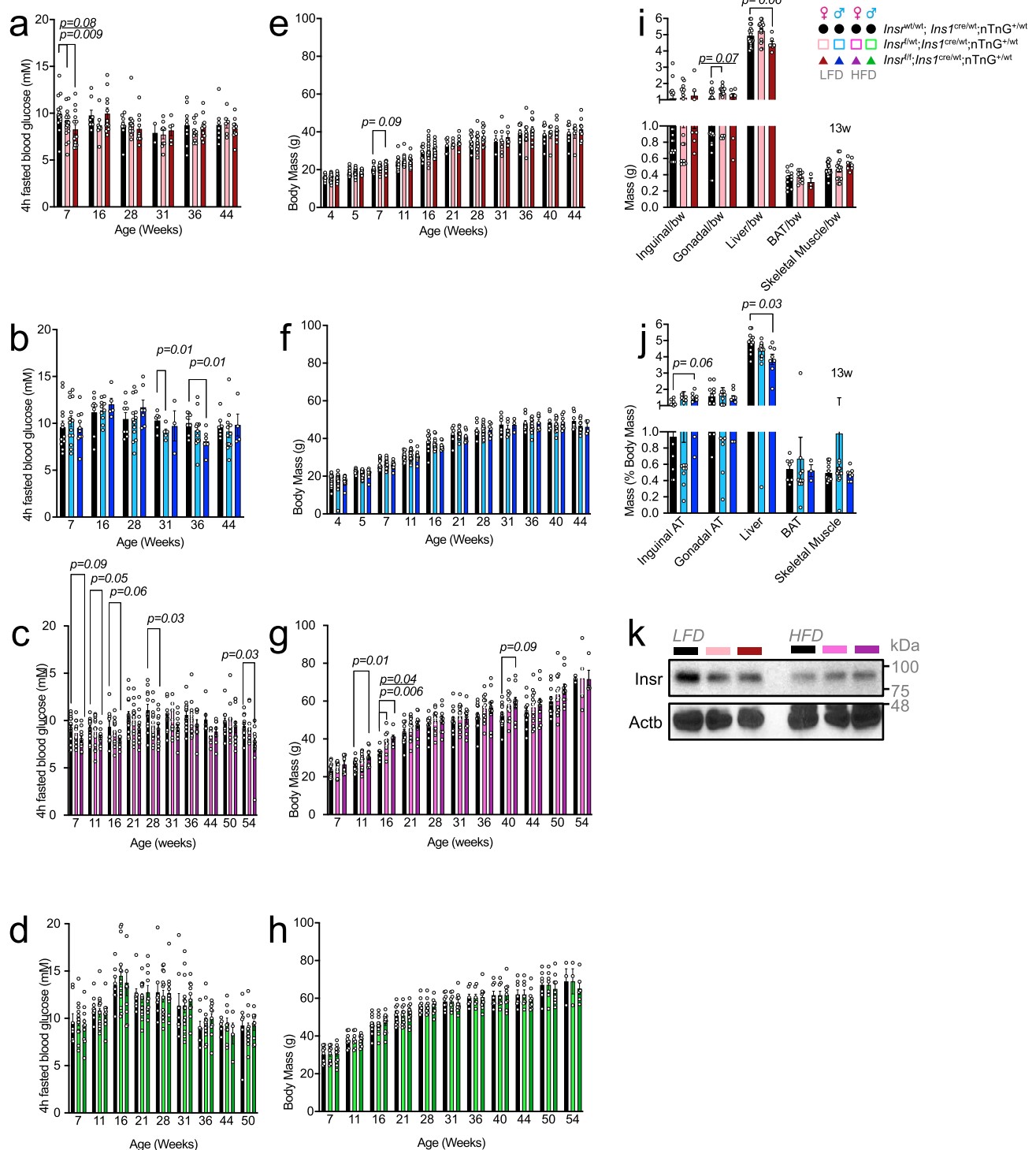

attributed to glucose in MIN6 cells require full insulin receptor expression[17]. We have previously found that glucose cannot stimulate primary mouse β-cell proliferation when autocrine insulin signaling is blocked by somatostatin and that "glucose-induced" Erk phosphorylation requires full insulin secretion[16,45]. On the other hand, inhibiting Insr in mouse islets with S961 or shRNA did not block glucose-induced proliferation of cultured β-cells[19]. Additional studies will be required to resolve this controversy.

Early hyperinsulinemia is a feature of β-cell *Insr* knockout models on multiple genetic backgrounds[20–22], including the

present study. Loss of Irs1 or Akt function results in basal hyperinsulinemia and, in some cases, increased β-cell mass[70,71], mimicking the early stages in the progression towards human type 2 diabetes. Our experiments begin to shed light on the systemic consequences of the hyperinsulinemia caused by β-cell-specific insulin resistance, which may be an early event in the pathogenesis of type 2 diabetes. Human data suggests that β-cell insulin resistance can be found in the obese state prior to hyperglycemia[8]. We and others have shown that hyperinsulinemia contributes to insulin resistance and obesity[14,39],

**Fig. 10 Effects of β-cell specific *Insr* deletion fasting glucose, body weight and organ weight. a–d** Plasma glucose concentration (mM) after a 4-h fast in control, βInsr$^{HET}$, and βInsr$^{KO}$ LFD-fed female (black, light pink, dark red bars; $n_{7week}$ = 15, 20, 14; $n_{16week}$ = 9, 7, 12; $n_{28week}$ = 9, 12, 10; $n_{31week}$ = 2, 9, 5; $n_{36week}$ = 9, 12, 10; $n_{44week}$ = 7, 5, 10) and male (black, light blue, dark blue bars; $n_{7week}$ = 15, 20, 14; $n_{16week}$ = 9, 7, 12; $n_{28week}$ = 9, 12, 10; $n_{31week}$ = 2, 9, 5; $n_{36week}$ = 9, 12, 10; $n_{44week}$ = 7, 6, 10) as well as HFD-fed female (black, bright pink, purple bars; $n_{7week}$ = 13, 12, 10; $n_{11week}$ = 13, 14, 12; $n_{16week}$ = 10, 13, 8; $n_{21week}$ = 14, 18, 12; $n_{28week}$ = 12, 14, 11; $n_{31week}$ = 12, 14, 12; $n_{36week}$ = 12, 13, 12; $n_{44week}$ = 6, 8, 7; $n_{50week}$ = 10, 12, 11; $n_{54week}$ = 10, 11, 11) and male (black, bright green, dark green bars; $n_{7week}$ = 8, 13, 10; $n_{11week}$ = 8, 15, 10; $n_{16week}$ = 8, 13, 10; $n_{21week}$ = 8, 15, 10; $n_{28week}$ = 8, 15, 10; $n_{31week}$ = 8, 14, 10; $n_{36week}$ = 7, 13, 10; $n_{44week}$ = 5, 9, 4; $n_{50week}$ = 6, 10, 9) mice. **e–h** Longitudinally tracked body mass (g) in control, βInsr$^{HET}$, and βInsr$^{KO}$ LFD-fed female (black, light pink, dark red bars; $n_{4week}$ = 18, 36, 14; $n_{7week}$ = 16, 18, 12; $n_{11week}$ = 19, 18, 12; $n_{16week}$ = 8, 20, 14; $n_{21week}$ = 6, 4, 7; $n_{28week}$ = 9, 10, 10; $n_{31week}$ = 3, 7, 5; $n_{36week}$ = 9, 12, 10; $n_{40week}$ = 7, 7, 9; $n_{44week}$ = 7, 6, 9) and male (black, light blue, dark blue bars; $n_{4week}$ = 19, 33, 17; $n_{7week}$ = 14, 22, 11; $n_{11week}$ = 13, 22, 10; $n_{16week}$ = 6, 14, 7; $n_{21week}$ = 6, 11, 7; $n_{28week}$ = 8, 13, 7; $n_{31week}$ = 8, 5, 3; $n_{36week}$ = 8, 13, 7; $n_{40week}$ = 8, 13, 7; $n_{44week}$ = 8, 13, 4) as well as HFD-fed female (black, bright pink, purple bars; $n_{7week}$ = 13, 12, 8; $n_{11week}$ = 17, 17, 12; $n_{16week}$ = 7, 9, 6; $n_{21week}$ = 12, 14, 10; $n_{28week}$ = 12, 13, 10; $n_{31week}$ = 12, 14, 10; $n_{36week}$ = 12, 13, 10; $n_{40week}$ = 11, 13, 10; $n_{44week}$ = 10, 13, 10; $n_{50week}$ = 10, 13, 10; $n_{54week}$ = 3, 4, 4) and male (black, bright green, dark green bars; $n_{7week}$ = 10, 12, 10; $n_{11week}$ = 8, 10, 8; $n_{16week}$ = 11, 14, 11; $n_{21week}$ = 11, 14, 11; $n_{28week}$ = 8, 10, 8; $n_{31week}$ = 8, 10, 8; $n_{36week}$ = 7, 10, 7; $n_{40week}$ = 8, 9, 8; $n_{44week}$ = 8, 9, 8; $n_{50week}$ = 7, 9, 7; $n_{54week}$ = 3, 6, 3) mice. **i, j** Weights, as a percentage of the whole body, of inguinal adipose tissue, gonadal adipose tissue, liver, brown adipose tissue and skeletal muscle. Data were analysed using repeated measures mixed effects models. All data were analysed by a fitted mixed-effects model with correction for multiple comparisons using Dunnett's method. **k** Representative immunoblot of Insr performed on mouse liver samples ($n$ = 3). All animals/samples were handled in a strictly blinded manner. All data are presented as mean values ± SEM. Source data are provided as Source Data file.

through multiple mechanisms including the downregulation of insulin receptors[40,72]. We observed propensity for excessive diet-induced body mass gain in mice with β-cell specific insulin resistance, as well as Insr protein downregulation in the liver. Thus, β-cell defects such as impaired autocrine feedback through Insr may contribute to insulin hypersecretion and accelerate the early stages of type 2 diabetes. In the later stages, the lack of pro-survival insulin signaling, perhaps in combination with other molecular defects, may contribute to failures in β-cell compensation and survival, thereby further accelerating the course of the disease[17,20,21,73,74].

Our investigation illustrates the power of using both females and males to study integrated physiology, although it was not designed to test specifically for sex differences. Indeed, β-cell *Insr* loss led to increased β-cell action potentials, calcium oscillations, and glucose-stimulated insulin secretion in 16-week-old female mice, but not in age-matched males. Although significant sex differences have been reported at the transcript level for many genes in mouse and human islets[75–78], no intrinsic sex differences of *Insr* mRNA levels were reported in sorted β-cells[77], and we confirmed this in our analysis of sorted β-cells (Fig. 2b). Given that we found identical *Insr* reduction in both sexes of βInsr$^{KO}$ mice, the sex differences in glucose-stimulated action potential firing rate, $Ca^{2+}$ oscillations, and insulin secretion must come from the sex-specific downstream transcriptomic consequences we identified (Fig. 2d–f). Of particular note, we identified a group of genes critical for mitochondrial ATP generation that were only significantly reduced in male β-cells, consistent with reduced ATP-coupled oxygen consumption at 10 mM glucose we measured in male, but not female, islets (Fig. 3k). This is likely to account for a least some of the observed sex difference. It is also possible that control male, but not female, β-cells were already maximally insulin resistant when studied. Given the abundance of data showing more pronounced insulin resistance in males[79,80], this could be another reason for female-specific response to β-cell *Insr* knockout. To this point, glucose tolerance is improved in 4-week-old βInsr$^{KO}$ males, an age at which control males remain insulin-sensitive, and in male mice with acute loss of *Insr*. More work will be needed to confirm this possibility, and to determine factors in addition to sex hormones and sex chromosomes that impact these sex differences in insulin sensitivity and glucose homeostasis. The sex-specific nature of phenotypes arising from our genetic manipulation of *Insr* in β-cells highlights the importance of including both sexes to accurately interpret data and to draw conclusions that will apply to both sexes. Additional

research will be required to fully elucidate the molecular mechanisms underlying the myriad of sex differences in β-cell physiology.

While our study is comprehensive and employs the best genetic tools available today, this work has limitations. *Ins1*$^{Cre}$ is the most β cell-specific Cre deletion strain available today[30], but this contention does not preclude off-tissue effects that have yet to be discovered. Cre recombinase itself is not totally benign. These facts and our detailed comparison of *Ins1*$^{wt/wt}$ mice with *Ins1*$^{Cre/wt}$ mice (Figs. S9–12), highlight the importance of the Cre control group we employed throughout our studies. Recently, it was reported that some colonies of *Ins1*$^{cre}$ mice exhibited silencing via DNA hypermethylation at the *Ins1* locus and this was suggested as an explanation for discordance between the phenotypes compared to gene deletions using Pdx1-Cre and Ins2-Cre transgenic lines[81]. In our study, we observed virtually complete recombination in β-cells and no evidence for off-tissue *Insr* deletion. We believe a major source of discrepancy with previously reported phenotypes stems from the propensity of previous promoter transgenic strains to recombine in the brain and robust expression of *Insr* throughout the brain. Another caveat of our experiments using βInsr$^{KO}$ mice is that *Insr* is expected to be deleted from β-cells starting in late fetal development[30]. In our hands, tamoxifen-inducible *Ins1*$^{CreERT}$ mice have insufficient β-cell specific recombination for in vivo physiological studies. Our validation experiments using the tamoxifen-inducible *Ins1*-CreERT address this limitation and confirm that β-cell insulin resistance improves glucose tolerance, at least under the initial insulin sensitive conditions. It should also be noted that, because we show that long-term deletion of *Insr* results in profound re-wiring of the β-cell transcriptome, the physiological changes can be the result of either direct or indirect action of *Insr* signaling. It should also be emphasized that, while we have deleted Insr, insulin can signal through Igf1r and Igf2r, especially at the higher concentrations predicted in the pancreas[44]. A further caveat is that the molecular mechanisms involved in insulin secretion may be different in mouse and human β-cells[82], although we note that the direction of effects we surmise agrees with the majority of human and canine studies, indicating general agreement across species[7,8,50,51].

In conclusion, our work demonstrates a context-dependent modulatory role for autocrine insulin negative feedback and the lack thereof (i.e., β-cell resistance) in insulin secretion, glucose homeostasis and body mass. We hope our studies help resolve longstanding and controversial questions about the local effect of

insulin on β-cells, and lead to experimental and theoretical studies that incorporate Insr-mediated signaling in other islet cell types.

## Methods

**Bioinformatics**. Human tissue-level proteome and transcriptome were downloaded from the ProteomicsDB resource (https://www.proteomicsdb.org)[83] in 2019 and sorted by relative protein abundance in Microsoft Excel. Publicly available human islet scRNAseq data were acquired from the panc8.SeuratData package and the SCTransform pipeline was followed to integrate the studies[84]. Expression data were normalized using the Seurat::NormalizeData function with default parameters and visualized using the Seurat::RidgePlot and Seurat::UMAPPlot functions, all from the Seurat package in R[85]. The volcano plot and the heatmap depicting the RNA sequencing data from mouse β-cells lacking Insr and littermate controls was created using the R packages EnhancedVolcano and gplots (heatmap.2 function), respectively.

**Mouse model and husbandry**. The in vivo experiments conducted in this current study were all in compliance with the ethical standards for animal investigation set by the Canadian Council on Animal Care. The current study received ethical approval from all involved institutions. Specifically, animal protocols were approved by the University of British Columbia Animal Care Committee (Protocol A16-0022), the Institutional Animal Care and Use Committee of the University of Massachusetts Medical School (A-1991-17) and the University of Michigan Institutional Animal Care and use Committee, in accordance with national and international animal care guidelines. Mice were housed at room temperature on a 12/12 light dark cycle at the UBC Modified Barrier Facility, unless otherwise indicated.

Whenever possible, we used both female and male mice for experiments, and at multiple ages. However, we could not always conduct the studies on both sexes side-by-side under the same conditions, and in some cases the group sizes are uneven between sexes and age-groups. Thus, although we can make confident statistically-backed conclusions about the role of Insr in both female and male mice, at the specific ages studied, our study was not designed to explicitly compare sex or age as biological variables.

Ins1^cre mice on a C57Bl/6 background (mix of N and J NNT alleles) were gifted to us by Jorge Ferrer[30] (now commercially available, Jax #026801). The Insr^f/f allele on a pure C57Bl/6J background (#006955) and the nuclear TdTomato-to-nuclear EGFP (nTnG) lineage tracing allele[32] on a mostly SV129 background with at least two backcrosses to C57Bl/6J before arriving at our colony were obtained from Jax (#023035) (Bar Harbor, ME). We generated two parental strains to avoid Cre effects during pregnancy; Ins1^cre/wt;Insr^f/wt male mice and Ins1^cre/wt;nTnG female mice. These two parental strains were crossed in order to generate full littermate insulin receptor knockout Insr^f/f;Ins1^cre/wt;nTnG (βInsr^KO) mice, partial insulin receptor knockout Insr^f/wt;Ins1^cre/wt;nTnG mice (βInsr^HET), and their control groups Insr^wt/wt;Ins1^cre/wt;nTnG mice (three alleles of insulin) and Insr^f/f;Ins1^wt/wt;nTnG mice (four alleles of insulin). Insr^f, Ins1^cre, and nTnG genotyping were done in accordance with Jax's recommendations using a ProFlex PCR system (Thermo Fisher Scientific, Canada). NNT genotyping was done as described previously[86]. Master mix for genotyping included 0.5 μM primers (Integrated DNA technologies), 2 mM dNTPs (New England Biolabs, #N0447S), 0.5U DreamTaq DNA polymerase (Fisher Scientific, #FEREP0702). Agarose gels varied from 1 to 2.5% (FroggaBio, #A87-500G). Primer sequences applied in this study: Insr forward: GAT GTG CAC CCC ATG TCT G; Insr reverse: CTG AAT AGC TGA GAC CAC AG; Ins1^cre forward: GGA AGC AGA ATT CCA GAT ACT TG; Ins1^cre/wt reverse: GTC AAA CAG CAT CTT TGT GGT C; Ins1^cre/wt mutual: GCT GGA AGA TGG CGA TTA GC; nTnG 9655: CCA GGC GGG CCA TTT ACC GTA AG; nTnG oIMR8545: AAA GTC GCT CTG AGT TGT TAT; nTnG oIMR8546: GGA GCG GGA GAA ATG GAT ATG; NNT (Wildtype) forward: GGGCATAGGAAGCAAATACCAAGTTG; NNT (Mutant) forward: GTGGAAT TCCGGTCGAGAGAACTCTT; NNT mutual reverse: GTAGGGCCAACTGTTTCT GCATGA. The mouse genetic background should be considered mixed with a majority of C57Bl6/J. The background was relatively fixed as we only employed littermate controls and backcrossed each mouse line every five generations.

In our studies, mice were fed 1 of 3 diets: either a chow diet (PicoLab Mouse Diet 20–5058) containing 23% protein, 22% fat, and 55% carbohydrates; a low-fat diet (LFD; Research Diets D12450B) containing 20% of kcal protein, 10% of kcal fat, and 70% of kcal carbohydrate including 35% sucrose, or; a high-fat diet (HFD; Research Diets D12492) containing 20% of kcal protein, 60% of kcal fat, and 20% of kcal carbohydrate including 10% sucrose. The mice in this study had ad lib access to water and food.

MIP^cre transgenic mice were obtained from Peter Dempsey at the University of Chicago[23]. The CAG-YFP reporter transgenic animals were from Jax (#011107). All animals in the MIP^cre study were males that were intraperitoneally injected at 8 weeks of age with tamoxifen (Sigma, T5648) freshly dissolved in corn oil (Sigma, C8267) with three injections at 200 mg/kg over a 1-week period.

**Comparison of control genotypes**. Before conducting our main study, we did a pilot experiment to determine whether the Ins1^cre knock-in mice had any

phenotype on their own under both low fat and high fat diets, and we tracked both "control" genotypes for the majority of our studies. Although Insr^wt/wt;Ins1^cre/wt;nTnG and Insr^f/f;Ins1^wt/wt;nTnG control mice exhibited generally similar phenotypes, we observed key differences that reinforced the rationale for using controls containing Cre and lacking 1 allele of Ins1, matching the experimental genotypes. For example, male HFD-fed Insr^f/f;Ins1^wt/wt;nTnG mice showed significantly higher levels of plasma proinsulin in comparison to Insr^wt/wt;Ins1^cre/wt;nTnG mice at 16 and 28 weeks of age (Fig. S7). At several ages, both LFD and HFD fed female Insr^f/f;Ins1^wt/wt;nTnG mice exhibited trends toward slightly improved glucose tolerance (Fig. S8), most likely due to one extra allele of insulin, in comparison to Insr^wt/wt;Ins1^cre/wt;nTnG mice. Insulin sensitivity was generally similar, although not identical (Fig. S9). Longitudinal tracking of body weight revealed a consistent tendency for mice with a full complement of insulin gene alleles to be heavier than mice in which 1 allele of Ins1 had been replaced with Cre. With the statistical power we had available, female HFD-fed Insr^f/f;Ins1^wt/wt;nTnG mice had significantly increased body mass at 11 and 16 weeks of age in comparison to Insr^wt/wt;Ins1^cre/wt;nTnG mice (Fig. S10). Once we had established the effects of the Ins1^cre allele on its own, we used a breeding strategy ensuring that all pups were born with three insulin alleles to control for any effects of reduced insulin gene dosage (see Fig. 1c). This strategy gave us cohorts of: Insr^wt/wt;Ins1^cre/wt;nTnG (βInsr^KO) control mice, Insr^f/wt;Ins1^cre/wt;nTnG (βInsr^KO) β-cell specific Insr heterozygous knockout mice, and Insr^f/f;Ins1^cre/wt;nTnG (βInsr^KO) β-cell specific Insr complete knockout mice. In some studies, the nTnG allele was not present (see Figure legends).

**Islet isolation and dispersion**. Mouse islet isolations were conducted by ductal inflation and incubation with collagenase, followed by filtration and hand-picking as in our previous studies[13,15,16,87] and following a protocol adapted from Salvalaggio[13,15,16,87]. Twenty-four hours of post-islets isolations, islets were washed (×4) (Ca/Mg-Free Minimal Essential Medium for suspension cultures, Cellgro #15-015-CV), followed by gentle trypsinization (0.01%), and resuspended in RPMI 1640 (Thermo Fisher Scientific #11875-093), 10% FBS, 1% PS. Cells were seeded either on glass cover slips or in 96-well plates according to the experimental procedure (see below).

**Immunoblotting**. Fifty islets per sample were washed in PBS twice and then lysed and sonicated in RIPA buffer (10 mM HEPES, 50 mM β-glycerol phosphate, 1% Triton X-100) supplemented with complete mini protease inhibitor cocktail (Roche, Laval, QC) and phosphatase inhibitors (2 mM EGTA, 70 mM NaCl, 347 1 mM Na$_3$VO$_4$, and 1 mM NaF). Protein concentration was measured using Micro BCA Protein Assay Kit (ThermoFischer Scientific). Ten microgram of total protein for each sample was separated by SDS-PAGE and transferred to immunoblot PVDF membrane (Bio-Rad Laboratories). Subsequently, membranes were blocked in I-Block (ThermoFischer Scientific) and probed with primary antibodies (see Supplement for list) targeting INSR-β subunit (1:1000, CST #3020S), ERK1/2 (1:1000, CST #4695), p-ERK1/2 (Thr202/Tyr204) (1:1000, CST #4370), AKT (1:1000, CST #9272), p-AKT (Thr308) (1:1000, CST #9275), ACTB (Novus Biologicals, NB600-501). Protein detection was performed by the use of the HRP-conjugated secondary antibodies: anti-rabbit (CST #7074) or anti-mouse (CST #7076) and Immobilon Forte Western HRP substrate (Millipore Sigma). Protein band intensity on exposed film was measured with Adobe Photoshop software version 7.

**Targeted gene expression analysis**. Tissue samples were kept frozen during grinding using mortals and pestles. cDNA was synthesized using qScript^TM cDNA synthesis kits (QuantaBio, #95047-500) following RNA isolation from 50 to 100 mg of sample using RNeasy mini kits (Qiagen, #74106) according to manufacturer's recommendations. qPCR was performed in 15 μl reaction volumes using a CFX384 Touch Real-Time PCR Detection System (BioRad). Primer sequences: Insr forward 5'- 376 TTTGTCATGGATGGAGGCTA-3' and Insr reverse 5'-CCTCATCTTGGGGTTGAACT-3'. Hprt forward 5'-TCAGTCAACGGGGGA CATAAA-3' and hprt reverse 5'-GGGGCTGT 379 ACTGCTTAACCAG-3'. Insr expression data were analyzed using the $2^{-\Delta\Delta CT}$ method using Hprt as a housekeeping gene. ΔCq = Cq(Insr)-Cq(Hprt) followed by normalization of the ΔCq(exp) to the mean of the Hprt expression in liver.

**RNA sequencing**. To generate transcriptomic data from β-cells lacking Insr and littermate controls, groups of 50 islets were dispersed using mild trypsin digestion according to our standard protocol[13,45], and FACS purified based on the GFP-positivity of the Ins1^Cre-induced nTnG allele. RNA isolation and library preparation were conducted in accordance with the SMART seq 2 protocol[88]. Sequencing of 100 GFP-positive β-cells was performed at the UBC Sequencing and Bioinformatics Consortium using Illumina NextSeq 500 with paired-end 75 bp reads. The number of read pairs per sample ranged from 4 million to 42 million, with a median of 18 million. Samples with high glucagon reads (>10,000 CPM) were omitted, as presumed β-cell purification failures. Raw counts of gene reads were quantified by Kallisto[89] and filtered for low count by only keeping genes with at least five counts in more than ten samples. Using DESeq2 package[90], gene counts were then normalized by variance stabilizing transformation and analyzed for

differential expression using Benjamini–Hochberg adjusted $p$ value <0.05 as the cut-off. Similar results were found when multiple analysis pipelines were applied and their results combined[91].

**Islet metabolism and oxygen consumption analysis**. The Seahorse XF Cell Mito Stress Test kit (cat#103015-100) was used to measure oxygen consumption rate (OCR) in dispersed mouse islets using an Agilent Seahorse XF96 Analyzer (Seahorse Bioscience, North Billerica, MA). Dispersed islet cells were seeded at a density of 40,000 cells/well in XF culture microplates. After seeding (48 h), the Seahorse XF Sensor Cartridge was hydrated in 180 μL of Seahorse XF Calibrant Solution (cat#100840-000) added to each well of the XF Utility Plate (cat#102416-100). The hydrated cartridge was kept in a non-CO2 incubator at 37 °C for 24 h to remove CO2 from media. To pre-equilibrate the cells, 180 μL of Seahorse XF base medium (minimal DMEM) containing 10 mM glucose, 4 mM L-glutamine, and 2 mM sodium pyruvate was added to each well of the culture plate 1 h prior to the run and was also present during extracellular flux measurements. Mitochondrial respiration was analyzed by sequential injections of modulators including oligomycin (2 μM) used to block ATP synthase, carbonyl-cyanide-4-(trifluoromethoxy) phenylhydrazone (FCCP 3 μM) to activate uncoupling of inner mitochondrial membrane allowing maximum electron flux through the electron transport chain, and finally a mix of rotenone (1 μM) and antimycin A (1 μM) to inhibit complexes I and III, respectively. Mitochondrial proton leak, ATP-linked oxygen consumption and non-mitochondrial respiration were calculated based on the resulting respiratory profile, as described[92].

**Patch-clamp electrophysiology**. Islets from 16-week-old chow-fed male and female mice were isolated at UBC. One hundred to three hundred islets from each mouse were shipped in a blinded manner overnight to the University of Alberta in RPMI (Invitrogen, 11875) with 10% FBS (Invitrogen #12483020), and 1% penicillin-streptomycin (Thermo Fisher, #15070063). Islets were dissociated into single cells using StemPro Accutase (Thermo Fisher Scientific, Cat# A11105-01) one day after receiving the islets. Dispersed cells were cultured in RPMI-1640 containing 11.1 mM glucose with 10% FBS and 100 U/ml penicillin/streptomycin for up to 2 days.

Membrane potential and current measurements were collected using a HEKA EPC10 amplifier and PatchMaster Software version 2 × 91(HEKA Instruments Inc, Lambrecht/Pfalz, Germany) in either the current-clamp or voltage-clamp mode in the perforated patch-clamp configuration. All the measures were done in a heated chamber (32–35 °C). Membrane potential measurement was performed with patch pipettes pulled from thick-walled borosilicate glass tubes (Sutter Instrument), with resistances of 8–10 MΩ when filled with 76 mM K2SO4, 10 mM KCl, 10 mM NaCl, 1 mM MgCl2 and 5 mM Hepes (pH 7.25 with KOH), and back-filled with 0.24 mg/ml amphotericin B (Sigma, cat# a9528). The extracellular solution consisted of 140 mM NaCl, 3.6 mM KCl, 1.5 mM CaCl2, 0.5 mM MgSO4, 10 mM Hepes, 0.5 mM NaH2PO4, 5 mM NaHCO3, 5 mM glucose (pH 7.3 with NaOH). Membrane potential was measured with 5 mM glucose starting from the beginning, for 5 min, then changed to 1 mM glucose for 4–5 min, then changed to 10 mM glucose for 8–10 min, finally changed back to 5 mM glucose. K$_{ATP}$ currents and reversal potential were recorded during and after membrane potential measurement on each cell. β-cells were distinguished by characteristic differences in the voltage-dependent inactivation of Na$^+$ channel currents[93].

Measurement of voltage-dependent exocytosis was performed in the whole-cell configuration. Before the start of whole-cell patch clamping, media were changed to bath solution containing (in mM): 118 NaCl, 20 Tetraethylammonium-Cl, 5.6 KCl, 1.2 MgCl2, 2.6 CaCl2, 5 HEPES, and 5 glucose (pH 7.4 with NaOH). For whole-cell patch-clamping, fire polished thin-walled borosilicate pipettes coated with Sylgard (3–5 MOhm), contained an intracellular solution with (in mM): 125 Cs-glutamate, 10 CsCl, 10 NaCl, 1 MgCl2, 0.05 EGTA, 5 HEPES, 0.1 cAMP, and 3 MgATP (pH 7.15 with CsOH). Quality control was assessed by the stability of seal (>10 GOhm) and access resistance (<15 MOhm) over the course of the experiment. Data were analysed using FitMaster (HEKA Instruments Inc) and Prism (GraphPad Software Inc., version 9, San Diego, CA).

**Calcium imaging and analysis**. Two days following cell seeding on glass coverslips, adherent islet cells were loaded with 5μM of the acetoxymethyl (AM) ester form of the calcium indicator Fura-2 (Thermo Fisher Scientific #F1221) for 30 min. Islet cells were perifused at 1 ml/min for 45 min prior to experimental procedure to ensure washout of excess FURA2. During experiments, cells were mounted on a temperature-controlled stage and held at 37 °C on a Zeiss Axiovert 200M inverted microscope equipped with a FLUAR 20× objective (Carl Zeiss, Thornwood, NY), while perifused with Krebs-Ringer (KRB) solution (144 mM NaCl, 5.5 mM KCL, 1 mM MgCl2, 2 mM CaCl2, 20 mM HEPES) of various glucose concentrations as indicated in figures. SlideBook 6 Digital microscopy software (Denver, CO) was applied.

Ca$^{2+}$ traces were analyzed automatically, as follows. Taking a similar approach to that described previously[34], eight features were extracted from the Traces (Fig. S2). Peaks during each phase were identified as local maxima reaching a value with a percent difference above the median baseline level greater than 20%. P-values for calcium analysis were generated using ANOVA with correction for multiple

comparisons performed using Tukey's method. Figures were generated using the ggplot2 package in R (Wickham[103]).

**Analysis of total protein synthesis rate**. For the purpose of pulse labeling of newly translated proteins, 50 isolated islets were incubated in complete RPMI media without cysteine and methionine (MP Biomedicals, #SKU 091646454) for 1 h. Subsequently media was supplemented with 250 μCi of [35S]-cysteine/methionine mixture (PerkinElmer, NEG772002MC) and islets were incubated under normal conditions for 30 min. Islets were then lysed and proteins separated by SDS-gel electrophoresis as described above. Gels were fixed for 30 min in 50% (v/v) ethanol in water with 10% (v/v) acetic acid, dried in gel dryer (Bio-Rad model 583) and then exposed to storage phosphor screen (GE Healthcare) overnight. Screens were imaged and digitized using Typhoon FLA 9000 biomolecular imager (GE Healthcare). Protein bands intensity was quantified with Adobe Photoshop software version 7.

**Dynamic insulin secretion perifusion analysis**. For perifusion experiments, islets from 16-week-old chow-fed male and female mice were isolated using collagenase, filtration and hand-picking as previously described[94]. Our standard approach compared the insulin response to both 20 mM and 10 mM glucose stimulation as well as direct depolarization with 30 mM KCl. More specifically, 150 hand-picked islets per column were perifused (0.4 ml/min) with 3 mM glucose KRB solution containing (in mM) 129 NaCl, 4.8KCL, 1.2 MgSO4•7H2O, 1.2 KH2PO4, 2.5 CaCl2, NaHCO3, 10 HEPES, as well as 0.5% BSA (Sigma # A7030) for 60 min to equilibrate the islets to the KRB and flow rate, and then treated as indicated. Samples were analyzed using a rat insulin radioimmunoassay that has 100% cross-reactivity for mouse insulin (Millipore-Sigma #ri-13k). Insulin content was measured after acid-ethanol extraction using an insulin ELISA (Stellux Rodent Insulin ELISA, Alpco #80-INSMR-CH10).

**Hyperglycemic clamps to assess glucose-stimulated insulin secretion in vivo**. In vivo hyperglycemic clamp experiments were performed at the National Mouse Metabolic Phenotyping Center (MMPC) at UMass Medical School. Body composition analysis was conducted by noninvasively measuring whole body fat mass and lean mass using $^1$H-MRS (Echo Medical Systems, Houston, TX). A survival surgery was performed at 5–6 days before hyperglycemic clamp experiments to establish an indwelling catheter in the jugular vein. On the day of experiment, mice were fasted overnight (~15 h), and a 2 h hyperglycemic clamp was conducted in awake mice by intravenously infusing 20% dextrose to rapidly raise and maintain plasma glucose levels at ~19 mM[95]. Blood samples were taken at 10–20 min intervals to measure plasma insulin levels during hyperglycemic clamps.

**Intravenous 4-day glucose infusion**. Mice were bred and genotyped at University of British Columbia and shipped at 5 weeks of age to the Division of Diabetes, Department of Medicine, University of Massachusetts Medical School, USA. In a blinded manner, glucose infusions were performed as described[96]. Jugular vein catheters were placed in 9–12-week-old male and female mice with blinded genotypes. From postoperative recovery through euthanasia mice were unrestrained and were maintained on a 12 h light/dark cycle, with access to 2.2 g diet (to ensure isocaloric intake across all mice) and water. After 2 days of recovery, mice received continuous 4-day intravenous infusions of 50% dextrose (Baxter) containing 500 μg/ml BrdU. Tail blood was sampled for plasma insulin, glucagon and blood glucose at Day 0, 1, 2, and 4. Blood glucose was measured using ReliOn glucometer (Walmart), glucagon was measured using mouse Glucagon ELISA (Mercodia 10-1281-01), and plasma insulin was measured using mouse Insulin ELISA kit (Mercodia 10-1247-01). Mice were euthanized at the end of the experiment and pancreas and duodenum were harvested for histology. Tissues were fixed for 5 h in 10% formalin and then stored in 1× PBS until processing, paraffin embedding and sectioning. Images were acquired using a NIKON fully motorized for Phase and Fluorescence Ti-E microscope. Images were taken of at least ten randomly selected islets, all four channels at the same time. To generate RGB images, channels were inserted to show Insulin-BrdU-DAPI, Glucagon-BrdU-DAPI or Insulin-Glucagon-DAPI. To generate yellow-magenta-white images to accommodate colorblind viewers, new files were generated in Adobe Photoshop in which original channel data were displayed in multiple channels using the merge function (e.g., to change green to yellow, green channel data were added to both green and red channels; for more detailed information please contact LCA). Cells were counted using Cell profiler automated counting software from Broad Institute (Cambridge, MA); all counts were manually checked.

**β-cell mass and immunohistochemistry**. Pancreata were perfused, then fixed for 24 h with 4% paraformaldehyde, and then washed twice with 70% ethanol prior to paraffin embedding and sectioning (5 μm) to obtain five different regions of the pancreas (100 μm apart) by WAXit Inc. (Vancouver, Canada). Paraffin was removed by 5 min xylene incubation steps. Sections were rehydrated in decreasing concentrations of ethanol and rinsed with water and PBS. Epitope retrieval was done either by immersing samples in 10 mM citrate buffer, pH 6.0 for 15 min at 95 °C, or by transferring sections to prewarmed 1 N HCl for 25 min at 37 °C. Samples were washed with PBS twice and were either blocked for 10 min at room

temperature (Dako protein block #X0909), or with goat block (GB) with Triton X-100 (10% BSA + 5% Goat Serum with 0.5% Triton X-100) for 1–4 h at room temperature. Samples were incubated overnight at 4 °C in primary antibodies targeting anti-insulin (1:200 Abcam #Ab7872), anti-glucagon (1:100 Cell Signaling Technologies, #2760S), anti-BrdU (1:250, Abcam ab6326), anti-GLUT2 (1:1000, Milipore, #07-1402). Following three PBS washes (5 min each), samples were incubated for 30 min or 1 h at room temperature in secondary antibodies in a light-deprived humid chamber. Secondary antibodies applied were anti-rabbit Alexa Fluor-488 (1:200, Invitrogen, # A-11008), anti-rabbit Alexa-488 (1:200, Invitrogen, #A11034), anti-rat Alexa-594 (1:200, Invitrogen, #A11007), anti-guinea pig Alexa-647 (1:200, Invitrogen, #A21450), anti-guinea pig Alexa-594 (1:200, Invitrogen #A-11076). Samples were mounted with either VECTASHIELD Hard Set Mounting Medium (Vector labs, # H-1500) or Fluoroshield both containing DAPI (Sigma-Aldrich, #F6182-20ML) following an additional three washes in PBS (10 min each). For β-cell area quantification, whole pancreas sections were imaged using an ImageXpress$^{MICRO}$ using a 10× (NA 0.3) objective and analyzed using the MetaXpress software (Molecular Devices, San Jose, CA, USA). Beta cell area was calculated as insulin positive area normalized to the entire pancreas of each section. The mean of five sections from five regions of the pancreas (100 μm apart) were quantified. For other immunofluorescence analysis of fixed tissue, we used a Zeiss 200 M microscope using 20× air objective (NA 0.75), NIKON fully motorized for Phase and Fluorescence Ti-E microscope. For live cell imaging for recombination validation, islets from $Ins1^{cre/wt}$:nTnG mice were incubated with CellMask™ Deep Red Plasma membrane stain (Thermo Fisher #C10046) using a Leica confocal microscope. For details on antibody validation please see validation statements on the manufacturer's websites collected in Table S2.

For β-cell mass analysis of *Insr* knockouts using MIP-Cre, β-cell mass determination was performed by intensity thresholding using the Fiji 2.1.0-/1.53c image analysis package, after fluorescence immunostaining for insulin on five independent sections per animal, selected at random throughout the pancreas as previously described. The proportion of insulin positive staining to total area was them multiplied by the pancreatic weight to derive the β-cell mass. The data was subsequently analyzed using a non-parametric statistical test (Mann–Whitney) in Prism 8.4.3 (GraphPad, San Diego, CA).

**Blood collection and in vivo analysis of glucose homeostasis and insulin secretion.** Tail blood was collected for blood glucose measurements using a glucometer (OneTouch Ultra 2 meter, Lifescan, Canada) for single time points as well as during glucose and insulin tolerance tests. Mice were fasted for 4 or 16 h during single timepoints and for 6 h during glucose and insulin tolerance tests, as well as glucose stimulated insulin secretion tests. The *i.p.* glucose dose was 2 g/kg unless otherwise specified. The *i.p.* Humalog (Eli Lilly and Co) insulin dose was 0.75U unless otherwise indicated. 1–2 days prior to femoral blood collection the experimental mice were tube handled and shaved. Femoral blood was collected for single timepoints, as well as for measurements of in vivo glucose-stimulated insulin secretion after *i.p.* injection of 2 g/ kg glucose. Blood samples were kept on ice during collection, centrifuged at 450 × *g* for 10 min at 4 °C and stored as plasma at −20 °C. Plasma samples were analysed for insulin (Stellux Chemi Rodent Insulin ELISA, Alpco #80-INSMR-CH10), proinsulin (Alpco #80-PINMS-E01), C-peptide (Alpco #80-CPTMS-E01), glucagon (Mercodia, #10-1281-01). Measurements were performed on a Spark plate reader (TECAN), and analysed using (GraphPad Software Inc., San Diego, CA).

**Mathematical modeling.** We used modified versions of the Topp model (Topp et al.[36]) to simulate glucose, insulin, and, in certain instances, β-cell dynamics:

$$\frac{dG}{dt} = R_0 - (E_{G0} + S_P I) G$$

$$\frac{dI}{dt} = \beta \frac{\sigma_0}{(1 + S_\beta I)} \cdot \frac{G^2}{(\alpha + G^2)} - kI$$

$$\frac{d\beta}{dt} = (-d_0 + r_1 G - r_2 G^2)\beta.$$

To capture the proposed insulin-receptor mediated negative feedback on insulin secretion, we introduced an inhibition factor, $1/(1 + S_\beta I)$, multiplying the insulin secretion term, parametrized by $S_\beta$, the β-cell Insr-specific insulin sensitivity. The other structural modification to the Topp model that we made was the removal of the β-cell mass equation when modeling glucose tolerance tests, justified by the slow nature of changes to β-cell mass compared to the fast changes in glucose and insulin levels during the test.

In glucose clamp conditions, a simple equation can be derived that relates $S_\beta$ to steady state insulin levels in wildtype and mutant mice: $S_\beta = (I_m^{ss} - I_{wt}^{ss})/(I_m^{ss})^2$, where $I_{wt}^{ss}$ is the steady state insulin level for wild type control mice, and $I_m^{ss}$ is the equivalent for the βInsrKO mice. We get this by simultaneously solving the steady state wt and mutant *I* equations for $S_\beta$. We assumed that, at each time point after the first one in the glucose-clamp insulin measurements, the blood insulin levels were hovering around steady state and all other parameters and variables were constant (Fig. S5). We used those insulin steady-state data points together with the simple equation to estimate $S_\beta$ for the male and female wild-type mice. The

equation for $S_\beta$ is parameter free, independent of the β-cell mass steady state, and depends only on the data so no parameter nor β-cell mass estimates were required.

For in silico glucose tolerance tests, we varied the values of $S_\beta$ and $S_P$ to see their influence on glucose homeostasis. To set the parameter values in our model for this exploration, we started with the Topp model, originally parametrized for humans and gradually varied them with two simultaneous objectives: (1) ensure that a simulated glucose tolerance test roughly matched a typical wildtype female low-fat diet glucose tolerance test time series and (2) the steady state β-cell mass of our full GIβ model roughly matched a typical observed β-cell mass in these mice. The resulting parameter values are given in Table S1 of the Supplemental Material. For in silico glucose tolerance tests, we then inserted the negative-feedback factor in the insulin secretion equation (referred to here as our full GIβ model or our fast GI model for the constant-β version) and used the parameters values estimated as described above. We held the β-cell mass constant at the steady state $\beta$ value of our full GIβ model, calculated using the clamp-data estimated $S_\beta$ value for wildtype female LFD mice and the $S_P$ value from the rough parameter estimation described above. For any individual tolerance test, we set $S_\beta$ and $S_P$, calculated the steady state of our fast GI model and used the steady state insulin value as the initial condition for insulin and the steady state glucose value plus 20 mM as the initial condition for glucose. Figure 6a shows sample in silico glucose tolerance tests for various values of $S_\beta$, with $S_P$ fixed at 70. The area under the glucose curve (AUGC) for the glucose tolerance test was then calculated for a range of $S_\beta$ and $S_P$ values, generating a numerical map from $S_\beta$ and $S_P$ to AUGC. From the glucose tolerance tests in wildtype and mutant male mice and wildtype and mutant female mice, we calculated the experimental AUGC values at 4, 9, 21, and 39 weeks. Using the computed AUGC map and the 11-week $S_\beta$ values calculated from the glucose-clamp data, we estimated the $S_P$ value that would give the experimental AUGC.

**Statistics.** Data are presented as mean ± SEM, with individual data points from biological replicates, unless otherwise indicated. Two-sided unpaired student's *t*-tests were performed when normally distributed data from only two groups were compared at a single timepoint. One-way ANOVA tests were applied when three groups were compared at a single timepoint. Mixed-effects model statistics (Prism, Graph Pad, version 9), which allows for missing measurements and un-even group sizes, were used for statistical analysis when two or more groups were compared at multiple time points (e.g., GTT, ITT, GSIS, 4 h fasted blood glucose, insulin, proinsulin, c-peptide and body weight). We corrected for multiple comparisons using Dunnett's method within experiments except when using Bonferroni adjusted *p* values in the RNA sequencing analysis.

For glucose tolerance test data, we also used a Bayesian approach with multilevel regression modeling due to the high number of correlated experiments with repeated measurements performed on the animals[97], in R version 4.1.0[98]. Missing values were multiply imputed using Amelia[99] with an *m* parameter of 10 if not missing at random (e.g., samples above or below a limit of detection) using priors of the limit of detection. Generally, a linear mixed-effects model was fitted with each animal treated as a random variable contributing to each measurement, and time treated as ordinal (minutes) or nominal (weeks). All Bayesian models were created in the Stan computational framework accessed with the brms package[100,101]. Posteriors were checked for goodness of fit and convergence. Default parameters for chain length, max_treedepth, and adapt_delta were used with 8000 iterations. Weakly informative priors were specified. If no result is reported, the posterior probability did not exceed 95%. Bayes factors are reported and strength of evidence assessed as per convention[102]. In related figures, lines are the fitted values of the posterior distribution with shading indicating the 95% credible interval. Related figures were generated using ggplot2[103]. Related tables were generated using gt package in R (https://CRAN.R-project.org/package=gt).

**Reporting summary.** Further information on research design is available in the Nature Research Reporting Summary linked to this article.

## Data availability

All data necessary to interpret and extend the research in of this study are available within this article, in the supplementary source data file. Source data for Figs. 1–10 are provided with this paper. Publicly available human islet scRNAseq data were acquired from the panc8.SeuratData package and the SCTransform pipeline was followed to integrate the studies[84]. The human islet scRNA, calcium, Bayesian IPGTT data source files generated in this study have been deposited in the github database and are available at [https://github.com/caraee/BIRKO]; [https://github.com/caraee/BIRKO_Ca], and [https://github.com/caraee/BIRKO] respectively. The Raw RNAseq data generated in this study have been deposited in the GEO database under accession code GSE159527. Source data are provided with this paper.

## Code availability

The codes generated to analyse the human islet scRNA, mouse RNA sequencing, calcium, mathematical model, and Bayesian IPGTT data have been deposited in the github database and are available at [https://github.com/caraee/BIRKO], [https://github.com/hcen/birko], [https://github.com/caraee/BIRKO_Ca], [https://github.com/

krfaulkner/beta-cell-insulin-resistance] and [https://github.com/caraee/BIRKO], respectively.

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

## Acknowledgements

S.S. was supported by an Alfred Benzon Post-Doctoral fellowship. Work was primarily supported by a CIHR grant (133692) to J.D.J. Work in Edmonton was supported by CIHR grant (148451) to P.E.M. Work in the Michigan Diabetes Research Center was supported by NIH (P30 DK020572). Work at U Mass Medical School was supported by NIH grants (5U2C-DK093000) to J.K.K. and (R01DK114686) to L.C.A. C.N. is supported as a Tier 1 CRC. Furthermore, we thank Johnson lab members for valuable feedback and those who provided insight when data were presented at conferences. We thank our animal care staff for supporting our animal husbandry, Bernard Thorens and Jorge Ferrer for sharing the *Ins1*cre knock-in mice, Eric Jan for allowing us to use his facilities to conduct S$^{35}$ labeling. We thank Quin Wills for helpful discussion of the smartseq2 protocol, and Austin Taylor for discussion of genotyping.

## Author contributions

S.S. led project design/management, conducted experiments, analyzed data, and wrote manuscript. E.P. conducted experiments, analyzed data, and edited manuscript (protein synthesis, Westerns). J.K. conducted experiments, analyzed data, and edited manuscript (ex vivo insulin secretion). H.H.C. analyzed data, and edited manuscript (RNAseq analysis). D.A.D. conducted experiments and analyzed data (in vivo glucose homeostasis). X-Q.D. conducted experiments, analyzed data, and edited manuscript (electrophysiology). R.B.S. conducted experiments, analyzed data, and edited manuscript (long-term hyperglycemia, proliferation). L.E. conducted experiments, analyzed data, and edited manuscript (MIP$^{Cre}$ mice). C.E. analyzed data, and edited manuscript (bioinformatics and calcium data analysis). K.F. analyzed data, and edited manuscript (mathematical modeling). S.A.M.M. conducted experiments and analyzed data (beta-cell area). N.N. conducted experiments (imaging). P.O. conducted experiments, analyzed data, and edited manuscript (Seahorse). D.H. conducted experiments (mRNA expression qPCR). X.H. conducted experiments (in vivo physiology). H.L. conducted experiments (in vivo physiology). H.M. conducted experiments, analyzed data, and edited manuscript (FACS, RNAseq prep). J.W. conducted experiments (beta cell INSR western blots). J.D.B. conducted experiments (liver INSR western blots). H.L.N. conducted experiments (hyperglycemic clamps). S.Su. conducted experiments (hyperglycemic clamps). A.B. conducted experiments (electrophysiology). R.K. conducted experiments (electrophysiology). B.G. conducted experiments (long-term hyperglycemia, proliferation). C.C-M. conducted experiments, and edited the manuscript (MIP$^{Cre}$ mice). S.F. analyzed data (RNAseq analysis). S.Si. conducted experiments (RNAseq). D.S.L. conducted experiments, analyzed data, and edited manuscript (Seahorse, nTnG validation). C.N. supervised

experiments, and edited manuscript (RNAseq). E.J.R. supervised experiments, and edited manuscript. E.N.C. supervised experiments, and edited manuscript (mathematical modeling). J.K. supervised studies, and edited manuscript (hyperglycemic clamps). E.B-M. supervised studies, and edited manuscript (MIP[Cre] mice). L.A. supervised studies, and edited manuscript (long-term hyperglycemia, proliferation). P.E.M. supervised studies, and edited manuscript (electrophysiology). J.D.J. conceived the project, oversaw its execution, edited manuscript, and guarantees the work.

## Competing interests
The authors declare no competing interests.

## Additional information

**Peer review information** *Nature Communications* thanks Roberto Bizzotto, Yasuo Terauchi and the other anonymous reviewer(s) for their contribution to the peer review this work. Peer reviewer reports are available.

Søs Skovsø[1], Evgeniy Panzhinskiy[1], Jelena Kolic[1], Haoning Howard Cen[1], Derek A. Dionne[1], Xiao-Qing Dai[2], Rohit B. Sharma[3], Lynda Elghazi[4], Cara E. Ellis[1], Katharine Faulkner[5], Stephanie A. M. Marcil[1], Peter Overby[1], Nilou Noursadeghi[1], Daria Hutchinson[1], Xiaoke Hu[1], Hong Li[1], Honey Modi[1], Jennifer S. Wildi[1], J. Diego Botezelli[1], Hye Lim Noh[6,13], Sujin Suk[6], Brian Gablaski[7,13], Austin Bautista[2], Ryekjang Kim[2], Corentin Cras-Méneur[8], Stephane Flibotte[9], Sunita Sinha[10], Dan S. Luciani[11], Corey Nislow[10], Elizabeth J. Rideout[1], Eric N. Cytrynbaum[5], Jason K. Kim[6,7], Ernesto Bernal-Mizrachi[12], Laura C. Alonso[3], Patrick E. MacDonald[2] & James D. Johnson[1]✉

[1]Diabetes Research Group, Life Sciences Institute, and Department of Cellular and Physiological Sciences, University of British Columbia, Vancouver, BC, Canada. [2]Alberta Diabetes Institute and Department of Pharmacology, University of Alberta, Edmonton, Canada. [3]Division of Endocrinology, Diabetes and Metabolism and the Weill Center for Metabolic Health, Weill Cornell Medicine, New York, NY, USA. [4]Department of Ophthalmology and Visual Sciences, University of Michigan Kellogg Eye Center, Ann Arbor, MI, USA. [5]Department of Mathematics, University of British Columbia, Vancouver, BC, Canada. [6]Program in Molecular Medicine University of Massachusetts Medical School, Worcester, MA, USA. [7]Division of Endocrinology, Diabetes and Metabolism, Department of Medicine, University of Massachusetts Medical School, Worcester, MA, USA. [8]Department of Internal Medicine, Division of Metabolism, Endocrinology and Diabetes, University of Michigan, Ann Arbor, MI, USA. [9]UBC Life Sciences Institute Bioinformatics Facility, University of British Columbia, Vancouver, BC, Canada. [10]UBC Sequencing and Bioinformatics Consortium, Pharmaceutical Sciences, University of British Columbia, Vancouver, BC, Canada. [11]BC Children's Hospital Research Institute, Department of Surgery, Faculty of Medicine, University of British Columbia, Vancouver, BC, Canada. [12]Division of Endocrinology, Diabetes and Metabolism, University of Miami Miller School of Medicine and Miami VA Health Care System, Miami, FL, USA. [13]Present address: Charles River Laboratories, Shrewsbury, MA, USA. ✉email: James.d.johnson@ubc.ca

