## [Peer Review File · Nature Communications]

Reviewers' Comments:

Reviewer #1:

Remarks to the Author:

In my initial critique of the manuscript "Beta-cell specific insulin resistance promotes glucose-stimulated insulin hypersecretion" NATMETAB-A20113918 by Skovsoe et al I stated "The concept addressing the mechanistic role of insulin signaling in the beta cell is important. However, this manuscript falls short in a clear way explaining the findings of the study and it is not clear to this reviewer what the main message is. As presented it is difficult to understand the importance of the findings." In their answer to my critique the authors now say that they have made significant changes to the manuscript to make it clearer. Nevertheless, I still think that the manuscript is not reader friendly and difficult to follow. There are a lot of data that are not fully explained. I am still confused about the gene data and what they really mean and I am lacking solid mechanistic data to explain the observed sex differences including hyperglycemia-induced proliferation of beta cells. It is good to have both sexes included but overall conclusions cannot just be made using female mice. Again in vitro beta cell action potentials, calcium oscillations and glucose stimulated insulin release are basic features and why should these be different in isolated beta cells from male or female mice. We need some more in-depth experiments to explain these differences What is for example the real differences in cytoplasmic ATP concentration, KATP channel activity and mitochondrial membrane potential between male and female Insr deleted beta cells. Hence, although addressing the mechanistic role of insulin signaling in the beta cell is important, I still think the revised version of the manuscript does not give a comprehensive and clear picture.

Reviewer #4:

Remarks to the Author:

I have carefully gone through the response and the new version of the manuscript. I am happy with the response and I believe this is important work in light of the controversy in this field. I have no further comments.

Reviewer #5:

Remarks to the Author:

The authors satisfactorily answered the reviewer questions on the mathematical modelling part of the paper. Two minor notes: the new Table S1 (mentioned in Materials and Methods) is very hard to find; Figure 6A still reports the G-I-beta equations, while the authors claimed that they replaced the G-I-beta equations in Fig 6A with the G-I equations.

The authors did not answer, instead, to the request for clarifications on apparent lack of correction of statistical significance levels for multiple inference. However, p value adjustment is reported in the text for gene expression, which is in the end the essential adjustment to perform.

Reviewer #6:

Remarks to the Author:

This manuscript investigated the metabolic effect of InsR deletion specific to pancreatic beta cells. The authors conducted an extensive analysis of autocrine beta insulin negative feedback in pancreatic beta cells, insulin secretion, glucose homeostasis and body mass. They demonstrated that beta cell InsR deletion increased β -cell action potentials, calcium oscillations, and glucose stimulated insulin secretion in 16-week-old female mice, but not in age-matched males. In the

revised version of the manuscript, they note that, in males only, a large number of mitochondrial genes, encoding proteins in oxidative phosphorylation Complex 1, Complex 3, and ATP-synthase that are down-regulated. This may account for significantly reduced ATP-coupled oxygen consumption seen in male, but not female.

Criticisms

1. Glucose tolerance is sometimes dependent on the mouse genetic background. Were all of the mice for experiments backcrossed to some background, such as C57bl/6?
2. As you pointed out, previous papers on *Insr* KO have reported that mice can become frankly diabetic. I agree with your opinion that reporting null effects or effects in the opposite direction are important. However, your conclusion in the Abstract is that loss of beta-cell *Insr* alone is sufficient to drive glucose-induced hyperinsulinemia, thereby improving glucose homeostasis. I am afraid to say that this statement is rather overstatement.

REVIEWER COMMENTS (our response in blue)

Reviewer #1:

In my initial critique of the manuscript "Beta-cell specific insulin resistance promotes glucose-stimulated insulin hypersecretion" NATMETAB-A20113918 by Skovsoe et al I stated "The concept addressing the mechanistic role of insulin signaling in the beta cell is important. However, this manuscript falls short in a clear way explaining the findings of the study and it is not clear to this reviewer what the main message is. As presented, it is difficult to understand the importance of the findings." In their answer to my critique the authors now say that they have made significant changes to the manuscript to make it clearer. Nevertheless, I still think that the manuscript is not reader friendly and difficult to follow. There are a lot of data that are not fully explained. Hence, although addressing the mechanistic role of insulin signaling in the beta cell is important, I still think the revised version of the manuscript does not give a comprehensive and clear picture.

- We acknowledge that this is a large and comprehensive paper, wherein we have been completely transparent with our data, and that some findings will require additional follow-up that is beyond the scope of our study. We are confident in the clarity of the central finding of our study, that loss of *Insr* increases beta-cell excitability, calcium oscillation frequency, and insulin secretion resulting in improved glucose tolerance in mice that are not already insulin resistant. The improved glucose tolerance is found in female mice on a low-fat diet, in young male mice on a low-fat diet, and reproduced after acute deletion in male mice of a complementary strain.
- We have also simplified the title somewhat.

I am still confused about the gene data and what they really mean and I am lacking solid mechanistic data to explain the observed sex differences including hyperglycemia-induced proliferation of beta cells. It is good to have both sexes included but overall conclusions cannot just be made using female mice. Again in vitro beta cell action potentials, calcium oscillations and glucose stimulated insulin release are basic features and why should these be different in isolated beta cells from male or female mice. We need some more in-depth experiments to explain these differences. What is for example the real differences in cytoplasmic ATP concentration, KATP channel activity and mitochondrial membrane potential between male and female *Insr* deleted beta cells.

- We respectfully disagree that more data are required to illustrate a sex difference. Our data clearly establish that sex differences in such 'basic features' are retained in isolated islets *ex vivo*. In fact, many other studies have found sex differences in fundamental aspects of beta-cell biology (we have added some of these new citations). Moreover, we have comprehensive unpublished data showing profound sex differences between isolated islets from male and female littermates (for example thousands of genes are significantly different in head-to-head comparison, either under basal or ER stressed conditions). We are writing our own separate, large dedicated paper to islets sex differences, but the published data and our own work in this manuscript are already sufficient to show clear sex differences.
 - As requested by the editor, we have now more clearly stated that the underlying mechanisms that explain these sex-differences are not well understood at this time and that further work is needed (end of the 3rd last paragraph in the Discussion). This included a clearer representation of the RNAseq data in the manuscript.

Reviewer #4:

I have carefully gone through the response and the new version of the manuscript. I am happy with the response and I believe this is important work in light of the controversy in this field. I have no further comments.

- We thank this reviewer for their positive assessment.

Reviewer #5:

The authors satisfactorily answered the reviewer questions on the mathematical modelling part of the paper. Two minor notes: the new Table S1 (mentioned in Materials and Methods) is very hard to find; Figure 6A still reports the G-I-beta equations, while the authors claimed that they replaced the G-I-beta equations in Fig 6A with the G-I equations.

- We thank the reviewer for pointing this out. We have fixed these issues, including by moving the equation details out of the Supplement into the main Figures.

The authors did not answer, instead, to the request for clarifications on apparent lack of correction of statistical significance levels for multiple inference. However, p value adjustment is reported in the text for gene expression, which is in the end the essential adjustment to perform.

- We apologize that this was not clear. We have now improved to the statistics part of the methods section to make clear the fact we have corrected for multiple comparisons in our statistics.
- Also, in response to this query, we have performed a Bayesian analysis of the glucose tolerance data (new Figure 8), which perhaps makes the differences between genotypes more clear.

Reviewer #6:

This manuscript investigated the metabolic effect of InsR deletion specific to pancreatic beta cells. The authors conducted an extensive analysis of autocrine beta insulin negative feedback in pancreatic beta cells, insulin secretion, glucose homeostasis and body mass. They demonstrated that beta cell InsR deletion increased β -cell action potentials, calcium oscillations, and glucose stimulated insulin secretion in 16-week-old female mice, but not in age-matched males. In the revised version of the manuscript, they note that, in males only, a large number of mitochondrial genes, encoding proteins in oxidative phosphorylation Complex 1, Complex 3, and ATP-synthase that are down-regulated. This may account for significantly reduced ATP-coupled oxygen consumption seen in male, but not female.

- We thank the reviewer for recognizing this interesting aspect of the complex phenotype of these mice.

Criticisms

1. Glucose tolerance is sometimes dependent on the mouse genetic background. Were all of the mice for experiments backcrossed to some background, such as C57bl/6?

- We thank the reviewer for bringing this up. These mice are on a mixed, but fixed background that is mostly C57Bl/6. We have now made clear the specific strains used in our crosses and the background considerations in the Methods section.

2. As you pointed out, previous papers on Insr KO have reported that mice can become frankly diabetic. I agree with your opinion that reporting null effects or effects in the opposite direction are important. However, your conclusion in the Abstract is that loss of beta-cell Insr alone is sufficient to drive glucose-induced hyperinsulinemia, thereby improving glucose homeostasis. I am afraid to say that this statement is rather overstatement.

- We thank the reviewer for this suggestion and have softened this statement in the Abstract (“contributing” rather than “sufficient” and “driving”).

Reviewers' Comments:

Reviewer #1:

Remarks to the Author:

Regarding the revision of manuscript "Beta-cell specific insulin resistance promotes glucose-stimulated insulin hypersecretion" NATMETAB-A20113918 by Skovsoe et al, I do not think that the authors have handled my critique in a satisfactory way. The manuscript remains not reader friendly in its present form and there are still data that are not fully explained. In this context the authors admit that some findings will require additional follow-up. My point is that if these data are presented within the manuscript they should be explained otherwise omitted.

Also, I do not dismiss that there are sex differences, but I would like to know more about why. Since the authors make a point of these sex differences they should also provide some sort of a mechanistic explanation. If the authors have some information regarding this interesting issue it should be included. It is not satisfactory to refer the readers to some unpublished data.

Overall, I think that the data in this manuscript are of potential interest. However, the authors should not just ignore my critique. An adequate handling of this critique should increase the quality and clarity of the manuscript and thereby make it publishable.

Reviewer #6:

Remarks to the Author:

I have no competing interests in relation to the paper which I am reviewing.

Whilst I did not see the first version of this paper, I was asked to comment on this revised version and assess the authors' response to Reviewer 3. I hear that this referee had a similar scientific background to mine, but unfortunately was unavailable to comment on this revised version.

I felt that the authors almost satisfactorily responded the criticisms raised by Reviewer 3 and therefore believed that this study was scientifically valid and technically sound. I felt, however, that conclusion in the Abstract that loss of beta-cell Insr alone is sufficient to drive glucose-induced hyperinsulinemia, thereby improving glucose homeostasis was rather overstatement. Now that the authors satisfactorily responded my criticisms, this version of the manuscript may be accepted for publication in Nature Communications.

Yasuo Terauchi

REVIEWERS' COMMENTS

Reviewer #1:

Regarding the revision of manuscript “Beta-cell specific insulin resistance promotes glucose-stimulated insulin hypersecretion” NATMETAB-A20113918 by Skovsoe et al, I do not think that the authors have handled my critique in a satisfactory way. The manuscript remains not reader friendly in its present form and there are still data that are not fully explained. In this context the authors admit that some findings will require additional follow-up. My point is that if these data are presented within the manuscript they should be explained otherwise omitted.

- We thank the reviewer for their perspective. However, we fundamentally disagree that all data presented in a paper must be ‘explained’. Indeed, this leads to an inevitable publication bias in only reporting some data. Here, we have made it clear which data we feel will require additional studies to fully understand.

Also, I do not dismiss that there are sex differences, but I would like to know more about why. Since the authors make a point of these sex differences they should also provide some sort of a mechanistic explanation. If the authors have some information regarding this interesting issue it should be included. It is not satisfactory to refer the readers to some unpublished data.

- We are delighted that the reviewer does not dismiss the existence of sex differences. We have provided RNAseq data that sheds light on the associated molecular mechanisms. As a courtesy to the reviewers, we revealed in our

previous response letter that a large manuscript focused entirely on sex differences is coming, but it is well beyond the scope of this study and not written yet.

Overall, I think that the data in this manuscript are of potential interest. However, the authors should not just ignore my critique. An adequate handling of this critique should increase the quality and clarity of the manuscript and thereby make it publishable.

- We look forward to the publication of our study and readers can decide whether it is of potential interest.

Reviewer #6

I felt that the authors almost satisfactorily responded the criticisms raised by Reviewer 3 and therefore believed that this study was scientifically valid and technically sound. I felt, however, that conclusion in the Abstract that loss of beta-cell Insr alone is sufficient to drive glucose-induced hyperinsulinemia, thereby improving glucose homeostasis was rather overstatement. Now that the authors satisfactorily responded my criticisms, this version of the manuscript may be accepted for publication in Nature Communications.

- We thank this reviewer for their helpful critique. As suggested, we softened the conclusion statement in the Abstract.